# Data-driven analysis to understand long COVID using electronic health records from the RECOVER initiative

Chengxi Zang[1], Yongkang Zhang[1], Jie Xu[2], Jiang Bian [2], Dmitry Morozyuk[1], Edward J. Schenck [3], Dhruv Khullar[1], Anna S. Nordvig[4], Elizabeth A. Shenkman [2], Russell L. Rothman[5], Jason P. Block[6], Kristin Lyman[7], Mark G. Weiner [1], Thomas W. Carton[7], Fei Wang [1] ✉ & Rainu Kaushal [1]

Recent studies have investigated post-acute sequelae of SARS-CoV-2 infection (PASC, or long COVID) using real-world patient data such as electronic health records (EHR). Prior studies have typically been conducted on patient cohorts with specific patient populations which makes their generalizability unclear. This study aims to characterize PASC using the EHR data warehouses from two large Patient-Centered Clinical Research Networks (PCORnet), INSIGHT and OneFlorida+, which include 11 million patients in New York City (NYC) area and 16.8 million patients in Florida respectively. With a high-throughput screening pipeline based on propensity score and inverse probability of treatment weighting, we identified a broad list of diagnoses and medications which exhibited significantly higher incidence risk for patients 30–180 days after the laboratory-confirmed SARS-CoV-2 infection compared to non-infected patients. We identified more PASC diagnoses in NYC than in Florida regarding our screening criteria, and conditions including dementia, hair loss, pressure ulcers, pulmonary fibrosis, dyspnea, pulmonary embolism, chest pain, abnormal heartbeat, malaise, and fatigue, were replicated across both cohorts. Our analyses highlight potentially heterogeneous risks of PASC in different populations.

The global COVID-19 pandemic from late 2019 has led to more than 620 million infections and 6.5 million deaths as of Oct 17, 2022[1]. Growing scientific and clinical evidence has demonstrated potential post-acute and long-term effects of SARS-CoV-2 infection in multiple organ systems[2], including cardiovascular[3], mental health[4], neurological[5], and metabolic[6] among other systems. Recently, several retrospective observational cohort analyses have described post-acute

sequelae of SARS-CoV-2 infection (PASC) using real-world patient data[7–9]. These studies typically start with a predefined list of PASC symptoms and signs and then contrast their incidence risk or burden in SARS-CoV-2 infected patients versus non-infected controls. Different analytical pipelines have been utilized, such as causal inference[7], regression analysis[10], and network analysis[11]. There are two major challenges to these existing studies. First, the disease etiology and

[1]Department of Population Health Sciences, Weill Cornell Medicine, New York, NY, USA. [2]Department of Health Outcomes Biomedical Informatics, University of Florida, Gainesville, FL, USA. [3]Department of Medicine, Division of Pulmonary and Critical Care Medicine, Weill Cornell Medicine, New York, NY, USA. [4]Department of Neurology, Weill Cornell Medicine, New York, NY, USA. [5]Center for Health Services Research, Vanderbilt University Medical Center, Nashville, TN, USA. [6]Department of Population Medicine, Harvard Pilgrim Health Care Institute, Harvard Medical School, Boston, MA, USA. [7]Louisiana Public Health Institute, New Orleans, LA, USA. ✉e-mail: few2001@med.cornell.edu

pathophysiology of PASC are complicated, and our current state of knowledge is still far from complete. This means that a conventional hypothesis-driven study design may miss potential PASC symptoms and signs. Second, prior studies have typically been conducted on one specific patient cohort without comparing different populations[1–3]. It is unclear how generalizable the results are from these studies when applied to the general patient population, and how PASC varies over broad patient populations with different characteristics.

In this study, we aim to address these two challenges by developing a high-throughput computational screening pipeline to identify potential PASC symptoms and signs using electronic health records (EHR) from two large Patient-Centered Clinical Research Networks (PCORnet)[12]: the INSIGHT network[13] covering patients in the New York City (NYC) metropolitan area and the OneFlorida+ network[14] covering patients from Florida. We started with a broad list of 137 potential PASC diagnoses and 459 potential PASC medications (See Method for the construction of both the diagnosis and medication lists). For each diagnosis or medication, we built an outcome-specific cohort with patients who were free of it at baseline, applied stabilized inverse probability of treatment weighting (IPTW) to adjust for high-dimensional hypothetical confounders collected from the baseline period, and calculated its adjusted hazard ratio and excess burden in the post-acute phase of the SARS-CoV-2 infection compared to non-infected patients (See an illustration in Fig. 1 and details in the Method section). We only focused on new incidences in the post-acute period in this study because it provided a clean way of defining PASC phenotypes without complicated consideration of pre-existing conditions. We found more PASC diagnoses and a higher risk of PASC in NYC than in Florida: 38 diagnosis categories and 59 medications involving a wide range of organ systems were identified to be associated with SARS-CoV-2 exposure from the INSIGHT cohort. However, by applying the same methodology for the OneFlorida+ cohort, we only found 11 diagnosis categories and 9 medications, and the majority of them were a subset of the findings from INSIGHT. The conditions replicated on both datasets were dementia, hair loss, pressure ulcers, pulmonary fibrosis, dyspnea, pulmonary embolism, chest pain, abnormal heartbeat, malaise, and fatigue, and diagnosis codes U099/B948 (Post COVID-19 condition, unspecified). These results highlighted the potential heterogeneity of PASC over different patient populations and the need for replication studies before robust conclusions about PASC can be made. This study is part of the NIH Researching COVID to Enhance Recovery (RECOVER) Initiative, which seeks to understand, treat, and prevent the post-acute sequelae of SARS-CoV-2 infection (PASC). For more information on RECOVER, visit https://recovercovid.org/.

## Results

### Population statistics
We identified potential PASC conditions using two different cohorts. The first cohort was built from the INSIGHT network[13], which contained 35,275 adult patients (age ≥ 20) with lab-confirmed SARS-CoV-2 infection who survived the first 30 days of infection from March 2020 to November 2021 in NYC and 326,126 eligible non-infected controls. Our second cohort was built from the OneFlorida+ network[14] with 22,341 eligible lab-confirmed SARS-CoV-2 positive patients who survived the first 30 days of infection during the same period in Florida, Georgia, and Alabama and 177,010 non-infected controls. To ensure that patients were connected to healthcare systems (and thus available for observation before and after their index encounters), we required eligible patients to have at least one diagnosis record within three years to one week before the index date and at least one diagnosis record within 30 days to 180 days after the index date. We also required no COVID-19-related diagnoses for the control patients (see Methods for the definitions of index date and lab confirmations and Fig. 1 for inclusion-exclusion cascade). We identified new-onset diagnoses and medications for SARS-CoV-2 infected patients in excess of

control patients 30–180 days after the index date as potential PASC conditions.

We summarized the baseline characteristics of both the INSIGHT cohort and OneFlorida+ cohort in Table 1 (See more characteristics in Supplementary Data 3) from information that was available on patients in clinical data; demographic information was collected from patients when they registered for care within the healthcare systems. We observed significant differences between the two cohorts regarding age, gender, race, area deprivation index, and outbreak waves. The INSIGHT cohort contained SARS-CoV-2 infected patients mainly from the New York metropolitan area with the median Area Deprivation Index (ADI, rankings from 1 to 100, with 1 and 100 indicating the lowest and highest level of disadvantage)[15] 15 (6–24) in the SARS-CoV-2 infected patient group, indicating fewer disadvantaged neighborhoods than the OneFlorida+ cohort whose median ADI was 58 (41–76). Indeed, the OneFlorida+ cohort consisted of a mixture of urban, suburban, and rural populations in Florida and selected cities in Georgia and Alabama (see Methods). The median age of SARS-CoV-2 infected patients in the INSIGHT cohort was 55 (38–68), older than the OneFlorida+ cohort with a median age of 50 (34–64). Plus, more female SARS-CoV-2 infected patients were in the OneFlorida+ cohort (62.7%) than in the INSIGHT cohort (58.6%). The INSIGHT cohort also had a more diverse population with 34.7% white and 54.9% others (Asian and others including American Indian or Alaska Native, Native Hawaiian or other Pacific Islander, multiple races, etc.); the OneFlorida+ cohort had a majority of patients identifying as White race (51.0%). Additionally, there is a higher proportion of patients infected early in the pandemic in the INSIGHT cohort (31.8% of all infected patients were from March 2020 to June 2020) compared to the OneFlorida+ cohort (9.1% of cases were from March 2020 to June 2020). Different temporal patterns of new cases per month across two cohorts are illustrated in Supplementary Fig. 1. The two networks also differed in care settings connected to patient encounters and treatments utilized for infected patients (e.g., more inpatient visits and more prescriptions of corticosteroids in the OneFlorida+ cohort than in the INSIGHT cohort).

### Results from the INSIGHT cohort
We started with a list of 137 potentially PASC-related diagnostic groups defined by ICD-10 diagnosis codes and CCSR categories (Supplementary Data 2) and 459 classes of medications grouped by their active ingredients (See Method) to screen for potential PASC conditions. For each of these diagnoses or medications, we built a condition-specific cohort in which patients didn't have the condition at baseline (Fig. 1) and conducted a stabilized inverse probability of treatment weighting for baseline covariates adjustment analysis following the pipeline detailed in the Method section and summarized in Supplementary Table 2 to estimate its incident risk in the post-acute period of SARS-CoV-2 infected patients compared to non-infected controls over 180 days. Figure 2 summarizes potential PASC diagnoses (Fig. 2a) and medications (Fig. 2b) identified from the INSIGHT cohort, spanning a broad range of organ systems. We reported incident risks in the adjusted hazard ratio (aHR) with 95% confidence intervals, along with adjusted cumulative incidence (CIF) per 1000 patients in two comparison groups.

**Nervous system.** We observed several neurologic conditions that exhibited higher risk in SARS-CoV-2 infected patients after acute infection, including myopathy (1.72 [95% CI, 1.32–2.24]), dementia (1.46 [95% CI, 1.29–1.65]), encephalopathy (1.46 [95% CI, 1.33–1.60]), cognitive problems (1.44 [95% CI, 1.35–1.55]), polyneuropathies (1.32 [95% CI, 1.16–1.51]), sleep disorders (1.28 [95% CI, 1.19–1.38]), headache (1.27 [95% CI, 1.17–1.37]), and anxiety (1.17 [95% CI, 1.09–1.26]). Besides, the insomnia drug melatonin also showed a significantly higher risk of use, in line with diagnoses of sleep disorders. Depression drugs quetiapine and mirtazapine also exhibited a higher risk of prescription.

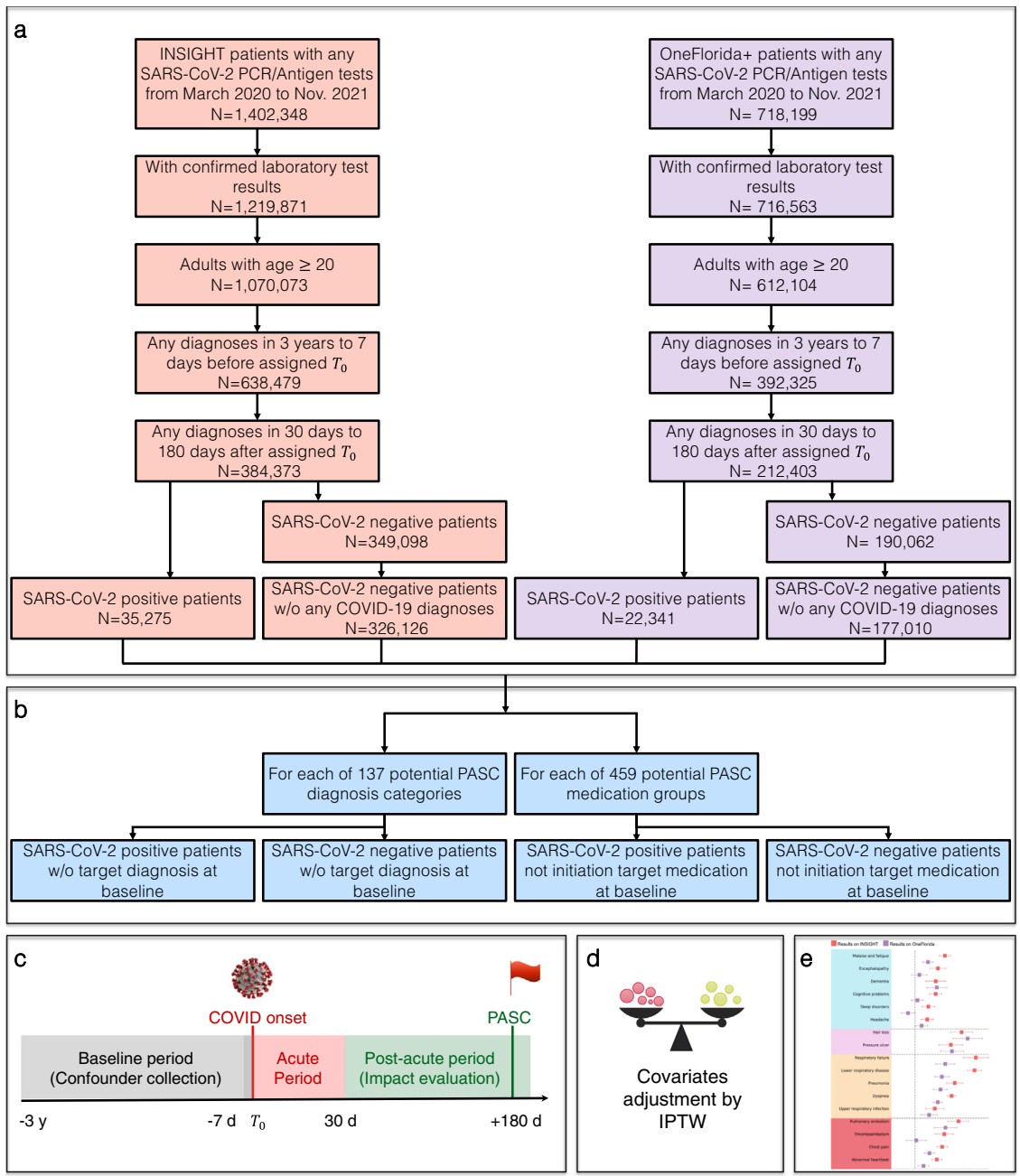

**Fig. 1 | Overall data-driven high-throughput screening framework. a** Selection of patients from the INSIGHT and OneFlorida+ EHR warehouses, March 2020 to November 2021. **b** High-throughput construction of PASC-specific case and control groups that patients did not have target condition at baseline. **c** Study design. The PASC outcomes were ascertained from day 30 after the SARS-CoV-2 infection and the adjusted risk was computed 180 days after the SARS-CoV-2 infection.

**d** Adjustment for baseline covariates by using stabilized inverse probability of treatment weighting (IPTW). **e** Likely PASC conditions were identified in the INSIGHT and OneFlorida+ cohorts respectively. Identified PASC were compared between the two cohorts. EHR electronic health records, PASC post-acute sequelae of SARS-CoV-2 infection.

**Skin.** Certain skin symptoms also showed significantly higher risk in the post-acute period, including hair loss (2.10 [95% CI, 1.84–2.39]), pressure ulcers (1.96 [95% CI, 1.70–2.27]), dermatitis (1.21 [95% CI, 1.10–1.33]) and paresthesia (1.17 [95% CI, 1.09–1.26]), coupled with relevant medications including witch hazel, collagenase, bacitracin, and loratadine.

**Respiratory system.** Several pulmonary manifestations in the post-acute phase were significant. These included pulmonary fibrosis (2.49 [95% CI, 2.29, 2.72]), dyspnea (1.80 [95% CI, 1.72, 1.89]), acute pharyngitis (1.41 [95% CI, 1.25–1.60]), chronic obstructive

pulmonary disease (COPD, 1.29 [95% CI, 1.15–1.44]), and atelectasis (1.27 [95% CI, 1.15–1.40]). Besides, a large number of medications in line with these diagnoses also showed significantly higher use, such as asthma or COPD drugs (e.g., vilanterol, fluticasone, budesonide, levalbuterol, formoterol, etc.) and cough suppressants (e.g., dextromethorphan, benzonatate, guaifenesin, etc.).

**Circulatory and blood.** Identified cardiovascular manifestations with a higher risk in the post-acute period were pulmonary embolism (2.25 [95% CI, 1.96–2.59]), thromboembolism (1.64 [95% CI,

**Table 1 | Baseline characteristics of the lab-confirmed SARS-CoV-2 positive patients and SARS-CoV-2 negative patients in the INSIGHT and OneFlorida+ cohorts, March 2020 to November 2021[a]**

| Characteristics | INSIGHT | | | OneFlorida+ | | |
|---|---|---|---|---|---|---|
| | SARS-CoV-2 Positive (N = 35,275) | SARS-CoV-2 Negative (N = 326,126) | SMD[b] | SARS-CoV-2 Positive (N = 22,341) | SARS-CoV-2 Negative (N = 177,010) | SMD[b] |
| Median age (IQR)—years | 55 (38–68) | 57 (40–69) | −0.09 | 50 (34–64) | 57 (40–69) | −0.27 |
| Age group—no. (%) | | | | | | |
| 20-<40 years | 9529 (27.0) | 77,403 (23.7) | 0.08 | 7506 (33.6) | 42,286 (23.9) | 0.22 |
| 40-<55 years | 7975 (22.6) | 70,313 (21.6) | 0.03 | 5473 (24.5) | 37,555 (21.2) | 0.08 |
| 55-<65 years | 6965 (19.7) | 66,361 (20.3) | −0.02 | 4036 (18.1) | 37,142 (21.0) | −0.07 |
| 65-<75 years | 5712 (16.2) | 62,860 (19.3) | −0.08 | 2929 (13.1) | 34,601 (19.5) | −0.17 |
| 75+ years | 5094 (14.4) | 49,189 (15.1) | −0.02 | 2397 (10.7) | 25,426 (14.4) | −0.11 |
| Sex—no. (%) Female | 20,686 (58.6) | 196,730 (60.3) | −0.03 | 14,004 (62.7) | 106,963 (60.4) | 0.05 |
| Male | 14,586 (41.3) | 129,360 (39.7) | 0.03 | 8335 (37.3) | 70,034 (39.6) | −0.05 |
| Race—no. (%) Asian | 1736 (4.9) | 17,439 (5.3) | −0.02 | 275 (1.2) | 2912 (1.6) | −0.03 |
| Black | 7791 (22.1) | 62,281 (19.1) | 0.07 | 6504 (29.1) | 35,381 (20.0) | 0.21 |
| White | 12,233 (34.7) | 139,512 (42.8) | −0.17 | 11,398 (51.0) | 105,521 (59.6) | −0.17 |
| Other | 9844 (27.9) | 69,406 (21.3) | 0.15 | 3730 (16.7) | 30,138 (17.0) | −0.01 |
| Missing | 3671 (10.4) | 37,488 (11.5) | −0.03 | 434 (1.9) | 3058 (1.7) | 0.02 |
| Ethnic group—no. (%) | | | | | | |
| Hispanic | 10,658 (30.2) | 73,522 (22.5) | 0.17 | 4500 (20.1) | 21,484 (12.1) | 0.22 |
| Not Hispanic | 20,838 (59.1) | 216,179 (66.3) | −0.15 | 14,798 (66.2) | 120,315 (68.0) | −0.04 |
| Unknown | 3779 (10.7) | 36,425 (11.2) | −0.01 | 3043 (13.6) | 35,211 (19.9) | −0.17 |
| Median ADI (IQR)—rank | 15 (6–24) | 13 (5–23) | 0.03 | 58 (41–76) | 53 (36–72) | 0.19 |
| BMI kg/m² (IQR) | 27 (21–32) | 25 (1–30) | 0.02 | 30 (25–35) | 28 (24–34) | 0.00 |
| Follow-up days (IQR) | 258 (163–418) | 269 (145–388) | 0.09 | 207 (109–367) | 250 (122–409) | −0.17 |
| Cares in the past 3 years—no. (%) | | | | | | |
| Inpatient 0 | 25,717 (72.9) | 278,784 (85.5) | −0.31 | 12,838 (57.5) | 112,480 (63.5) | −0.12 |
| Inpatient 1–2 | 6805 (19.3) | 37,297 (11.4) | 0.22 | 4614 (20.7) | 33,658 (19.0) | 0.04 |
| Inpatient >=3 | 2753 (7.8) | 10,045 (3.1) | 0.21 | 4889 (21.9) | 30,872 (17.4) | 0.11 |
| Corticosteroids Prescription | 4999 (14.2) | 28,915 (8.9) | 0.17 | 4253 (19.0) | 27,783 (15.7) | 0.09 |
| Immunosuppressant Prescriptions | 2110 (6.0) | 10,761 (3.3) | 0.13 | 1013 (4.5) | 7281 (4.1) | 0.02 |
| Index time—no. (%) | | | | | | |
| 03/20-06/20 | 11,235 (31.8) | 53,988 (16.6) | 0.36 | 2032 (9.1) | 37,363 (21.1) | −0.34 |
| 07/20-10/20 | 2018 (5.7) | 111,409 (34.2) | −0.76 | 6035 (27.0) | 54,060 (30.5) | −0.08 |
| 11/20-02/21 | 14,637 (41.5) | 88,009 (27.0) | 0.31 | 6254 (28.0) | 38,536 (21.8) | 0.14 |
| 03/21-06/21 | 5573 (15.8) | 54,234 (16.6) | −0.02 | 2315 (10.4) | 27,985 (15.8) | −0.16 |
| 07/21-11/21 | 1812 (5.1) | 18,486 (5.7) | −0.02 | 5705 (25.5) | 19,066 (10.8) | 0.39 |

[a]The lab-confirmed SARS-CoV-2 positive and negative patients were identified by polymerase chain reaction (PCR) test or antigen test. Negative patients have further required no documented COVID-19-related diagnoses at baseline. IQR denotes the interquartile range. The percentage may not sum up to 100 because of rounding. ADI, the Area Deprivation Index. BMI, Body Mass Index.
[b]A standardized mean difference (SMD) of >0.10 or <−0.10 indicates an important effect size difference between the two samples, otherwise, no significant difference is assumed.

1.47–1.84]), chest pain (1.55 [95% CI, 1.46–1.66]), abnormal heartbeat (1.40 [95% CI, 1.32–1.49]), hypotension (1.34 [95% CI, 1.20–1.50]), and heart failure (1.23 [95% CI, 1.12–1.35]), coupled with anticoagulant medications (e.g., apixaban, rivaroxaban, enoxaparin, heparin, etc.) and beta-blocker metoprolol. We also observed a higher risk of anemia (1.32 [95% CI, 1.24–1.41]) and ferric cation use in the post-acute phase.

**Endocrine.** Identified endocrine, nutritional, and metabolic disorders with higher risk were malnutrition (1.57 [95% CI, 1.43–1.72]), fluid and electrolyte disorders (1.32 [95% CI, 1.23–1.41]), diabetes mellitus (1.27 [95% CI, 1.17–1.37]), and edema (1.23 [95% CI, 1.16–1.30]), coupled with higher use of glucagon, insulin, and metformin.

**Digestive system.** Digestive system conditions with higher risks were constipation (1.19 [95% CI, 1.11–1.28]) and abdominal pain (1.18 [95% CI,

1.12–1.24]). The associated medications included magnesium hydroxide, esomeprazole, and simethicone.

**Genitourinary system.** Cystitis (1.31 [95% CI, 1.15–1.49]) and acute kidney failure (1.25 [95% CI, 1.15–1.36]), and associated tamsulosin, showed higher risk in the post-acute period.

**General or Musculoskeletal.** General symptoms include malaise and fatigue (1.64 [95% CI, 1.54–1.75]), fever (1.49 [95% CI 1.34–1.66]), dizziness (1.24 [95% CI 1.13–1.36]), joint pain (1.18 [95% CI, 1.12–1.24]), and fibromyalgia (1.18 [95% CI 1.09–1.27]) showed significantly higher risk, coupled with a higher risk of using ibuprofen, ketorolac, and acetaminophen. In addition, we also investigated two ICD-10 diagnosis codes B948 (sequelae of other specified infectious and parasitic diseases) and U099 (post-COVID-19 condition, unspecified), which showed significantly higher risk (42.7 [95% CI, 28.9 63.2]).

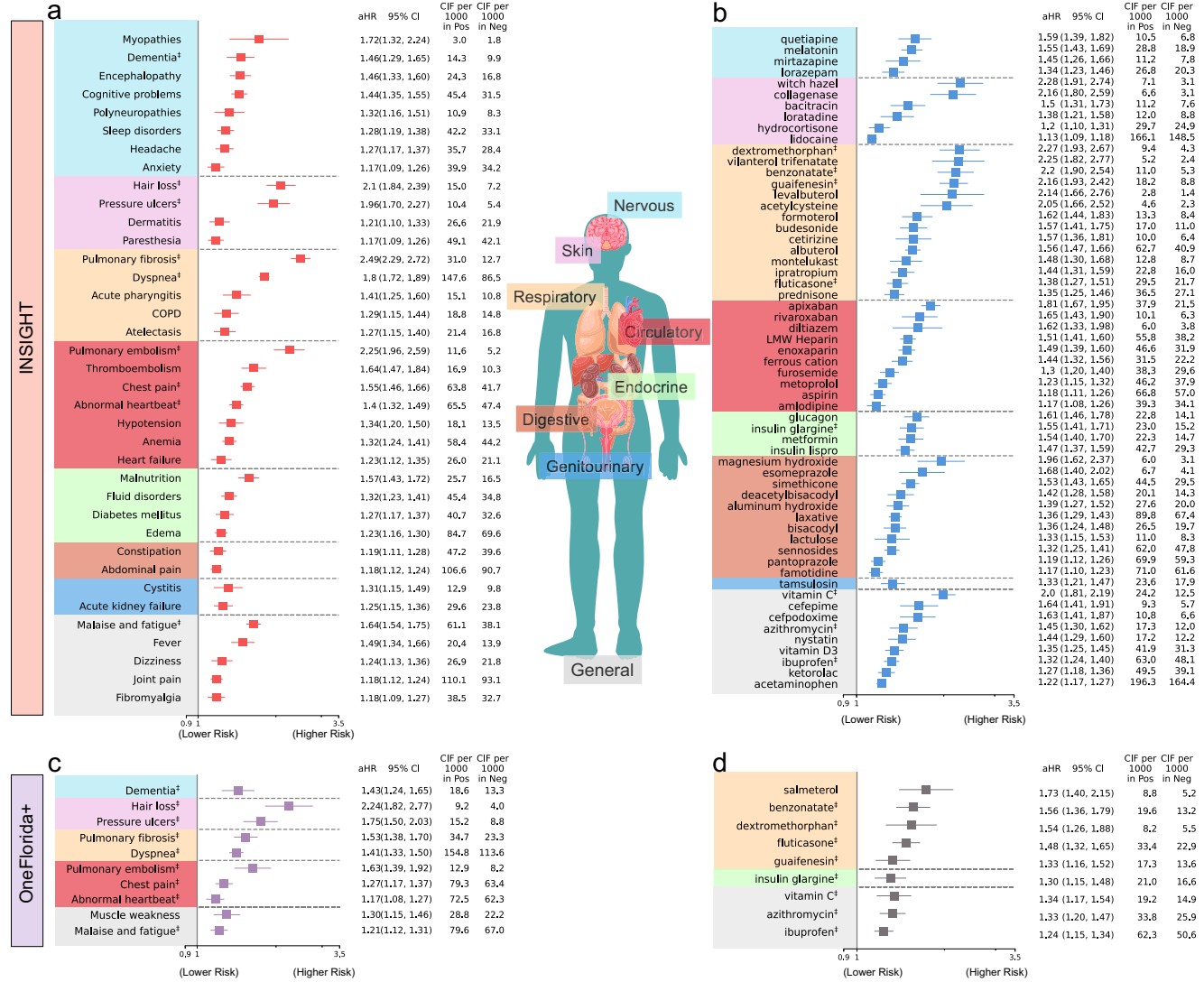

**Fig. 2 | Identified potential incident PASC conditions from the INSIGHT cohort and the OneFlorida+ cohort, March 2020 to November 2021. a** The risk of incident diagnoses from INSIGHT. **b** The risk of incident medications from INSIGHT. **c** The risk of incident diagnoses from OneFlorida+. **d** The risk of incident medications from OneFlorida+. The incident risk was quantified by the adjusted hazard ratios with 95% confidence intervals, and we also reported the adjusted cumulative incidences per 1000 patients in both the SARS-CoV-2 positive and the negative groups. The sequelae outcomes were ascertained from day 30 after the SARS-CoV-2 infection and computed 180 days after the SARS-CoV-2 infection. PASC conditions were selected based on adjusted hazard ratio > 1, the aHR's *P*-value

$< 8.39 \times 10^{-5}$ (the Bonferroni-corrected significance threshold for multiple comparisons), and at least 100 identified cases in the positive group. The aHR and its *P*-value were calculated by the Cox proportional hazard model and the Wald Chi-Square test. The colors represent different organ systems. The replicated diagnoses and medications in the OneFlorida+ were marked by ‡ symbols. aHR adjusted hazard ratio, CI confidence interval, CIF adjusted cumulative incidence function, COPD Chronic obstructive pulmonary disease. The PASC diagnosis code U099/B948 was also replicated in both cohorts but not illustrated in **a** or **c**, with aHRs 42.7 (95% CI, 28.9–63.2) and 39.8 (95% CI, 26.8–59.0) for INSIGHT and OneFlorida+, respectively.

## Results from the OneFlorida+ cohort and comparison with INSIGHT

To better understand the heterogeneity and commonality of potential PASC conditions over different populations, we replicated our analysis on the OneFlorida+ cohort and compared the PASC risks in OneFlorida+ versus INSIGHT. We summarized identified diagnoses (Fig. 2c) and medications (Fig. 2d) from the OneFlorida+ cohort and further compared these incident diagnoses (Fig. 3) and medications (Supplementary Fig. 3) with INSIGHT. The replicated conditions in both cohorts were highlighted by ‡ symbols.

As shown in Fig. 2c, d, 11 PASC diagnostic conditions (including U099/B948) and 9 medications were identified as significant in the OneFlorida+, which were fewer than INSIGHT when using the same screening criteria (method section). As shown in Fig. 3, the overall adjusted hazard ratios were lower in the OneFlorida+ cohort than in

the INSIGHT cohort, indicating a generally lower relative risk of potential PASC conditions in OneFlorida+ than in INSIGHT. For certain PASC conditions, the associated aHR values in the INSIGHT cohort exceed that in the OneFlorida+ cohort by more than 30%, such as myopathy, encephalopathy, sleep disorders, anxiety, pulmonary fibrosis, thromboembolism, anemia, heart failure, malnutrition, malaise, and fatigue.

**Nervous system.** The neurologic condition also replicated from the OneFlorida+ in the post-acute phase was dementia which showed aHR 1.43 [95% CI, 1.24–1.65] in OneFlorida+ versus 1.46 [95% CI, 1.29–1.65] in INSIGHT.

**Skin.** The replicated skin symptoms included hair loss (2.24 [95% CI, 1.82–2.77] OneFlorida+ vs. 2.10 [95% CI, 1.84–2.39] INSIGHT) and

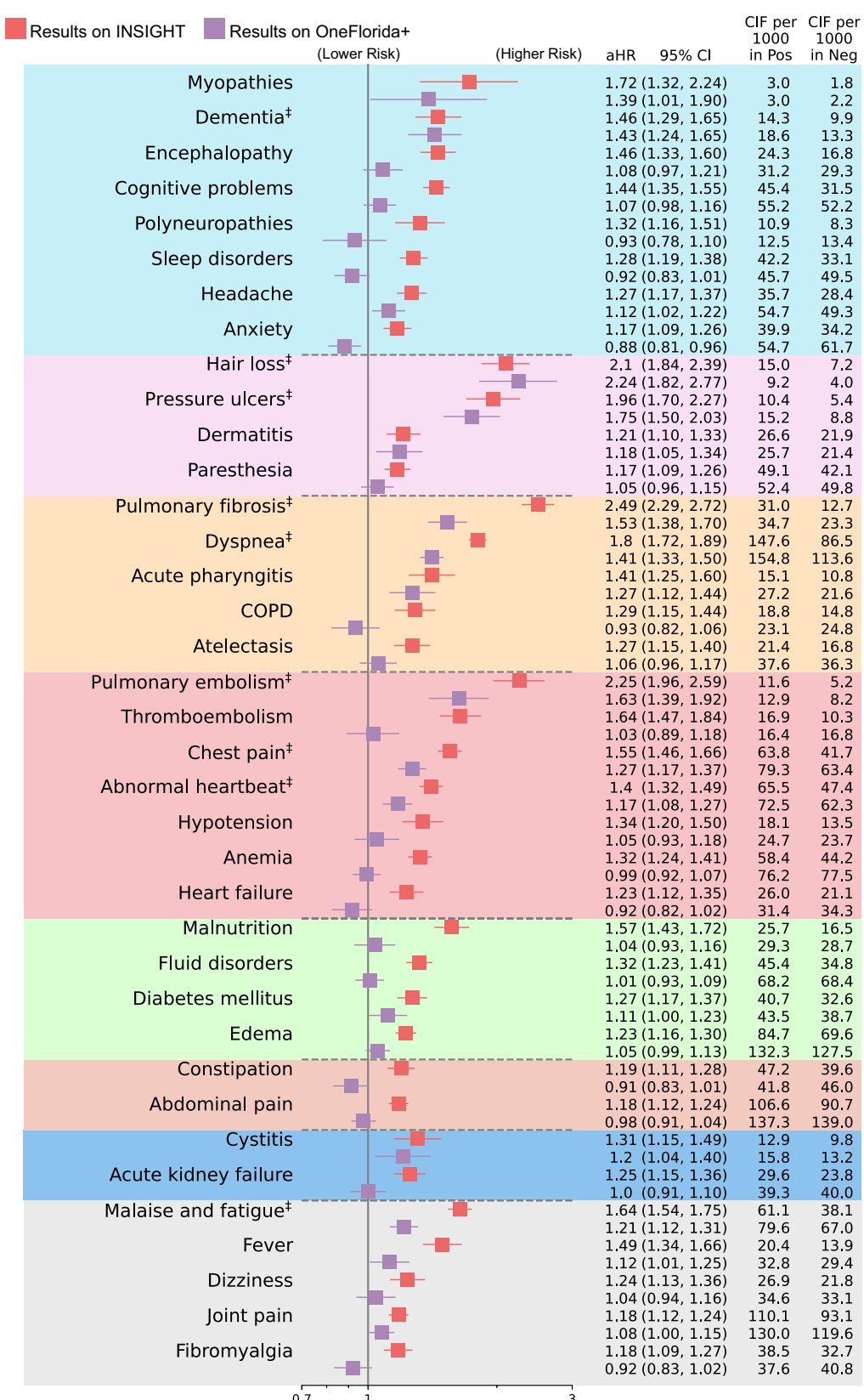

pressure ulcers (1.75 [95% CI, 1.50–2.03] OneFlorida+ vs. 1.96 [95% CI, 1.70–2.27] INSIGHT).

**Respiratory system.** The replicated pulmonary manifestations were pulmonary fibrosis (1.53 [95% CI, 1.38–1.70] OneFlorida+ vs. 2.49 [95% CI, 2.29, 2.72] INSIGHT) and dyspnea (1.41 [95% CI, 1.33, 1.50] OneFlorida+ vs.1.80 [95% CI, 1.72, 1.89] INSIGHT). The replicated medications included asthma or COPD drugs fluticasone, and cough suppressants dextromethorphan, benzonatate, and guaifenesin (Supplementary Fig. 3).

**Fig. 3 | Comparison of the PASC risks in the INSIGHT cohort versus in the OneFlorida+ cohort, from March 2020 to November 2021.** The incident risk was measured by the adjusted hazard ratios (aHR) with 95% confidence intervals as shown in the main panel. The adjusted cumulative incidences (CIF) per 1000 patients in both the SARS-CoV-2 positive group and the negative group were also reported. The PASC conditions identified in both datasets were marked by ‡ symbols. The color panels represent different organ systems, including (from top to bottom): the nervous system, skin, respiratory system, circulatory system, endocrine and metabolic, digestive system, genitourinary system, and other signs. The PASC outcomes were ascertained from day 30 after the SARS-CoV-2 infection and all the adjusted risk measures were computed 180 days after the SARS-CoV-2 infection. The aHRs of PASC diagnosis code U099/B948 were not illustrated here. COPD, Chronic obstructive pulmonary disease. PASC, post-acute sequelae of SARS-CoV-2 infection.

**Circulatory and blood.** The replicated cardiovascular conditions with a higher risk in the post-acute period were pulmonary embolism (1.63 [95% CI, 1.39–1.92] OneFlorida+ vs. 2.25 [95% CI, 1.96–2.59] INSIGHT), chest pain (1.27 [95% CI, 1.17–1.37] OneFlorida+ vs. 1.55 [95% CI, 1.46–1.66] INSIGHT), and abnormal heartbeat (1.17 [95% CI, 1.08–1.27] OneFlorida+ vs. 1.40 [95% CI, 1.32–1.49] INSIGHT).

**Endocrine.** We observed replicated insulin use in both datasets, e.g., insulin glargine with aHR 1.30 [95% CI, 1.15–1.48] in OneFlorida+ versus 1.55 [95% CI, 1.41–1.71] in INSIGHT.

**General or musculoskeletal.** The replicated general symptoms include malaise and fatigue (1.21 [95% CI, 1.12–1.31] OneFlorida+ vs. 1.64 [95% CI, 1.54–1.75] INSIGHT), and PASC diagnosis code U099/B948 (39.8 [95% CI, 26.8 59.0] OneFlorida+ vs. 42.7 [95% CI, 28.9 63.2] INSIGHT). The replicated medications included vitamin C, ibuprofen, and azithromycin.

## Stratified analysis

To further understand the heterogeneity of potential PASC conditions over different subgroups, we conducted a comprehensive stratified analysis to examine how identified diagnoses vary across demographic (age, gender, race) groups, baseline pre-existing conditions, disease severity in the acute phase according to healthcare utilizations (outpatient versus inpatient), and different infection waves, on both the INSIGHT and OneFlorida+ cohorts. We also studied the subpopulation that had no documented comorbidities (a tailored list of the Elixhauser comorbidities[16] and related drug categories, details provided in the Method section) or PASC-like symptoms (our PASC diagnosis screening list) at baseline, referred to as the healthy population.

For each subgroup analysis, we built the infected subpopulation and its control subpopulation and re-estimated the stabilized IPTW weights for adjustment. Here we quantified PASC risk by the excess burden per 1000 patients[17], which is defined as the difference between the adjusted cumulative incidences (at 180 days after baseline) of a specific condition in the SARS-CoV-2 infected patient subpopulation versus the control subpopulation. We considered death after 30 days after the infection as a competing risk. Figure 4 and Supplementary Fig. 2 summarized the subgroup excess burden of PASC in INSIGHT and OneFlorida+ respectively, which are further described below. See the adjusted cumulative incidence of PASC diagnoses in the infected group in Supplementary Fig. 4 and 5. Of note, all the following subgroup analysis results were interpreted in terms of adjusted excess burdens (See the stratified results in terms of adjusted hazard ratios in Supplementary Fig. 12 and 13).

**Acute phase severity.** General respiratory symptoms and signs (Fig. 3) demonstrated increasing burdens by settings (e.g., dyspnea from 40.1 excess cases per 1000 patients compared to control patients in the outpatient setting to 89.9 in the inpatient setting). Other potential PASC diagnoses that followed the same trend included pulmonary fibrosis (7.8–34.1), dementia (2.3–5.1), hair loss (6.4–11.5), pulmonary embolism (2.7–10.6), chest pain (19.2–24.4), abnormal heartbeat (14.7–19.9), and malaise and fatigue (15.6–27.4). We further investigated two PASC-related

ICD-10 diagnosis codes, U099 (post-COVID-19 condition, unspecified) and B948 (sequelae of other specified infectious and parasitic diseases), which also showed an increasing burden from 3.9 in the outpatient setting to 14.2 in the inpatient setting. All the above-mentioned trends were further replicated in the OneFlorida+ cohort (Supplementary Fig. 2).

**Age groups.** We partitioned patients into two groups according to their age (<65 and ≥ 65). Potential PASC conditions that had the highest excess burden in <65 groups were dyspnea, chest pain, abnormal heartbeat, malaise, and fatigue in both cohorts. Potential PASC conditions with the highest excess burden in the ≥ 65 groups included dyspnea, pulmonary fibrosis, dementia, pressure ulcers, pulmonary embolism, malaise, and fatigue, among others; all of these conditions had a higher excess burden among those ≥65 compared to <65 in both the cohorts.

**Gender and race.** Higher excess burdens in male patients included dyspnea, pulmonary fibrosis, chest pain, malaise, and fatigue in both two cohorts. Problems including hair loss demonstrated higher excess burdens for female patients. Black patients had higher excess burdens of chest pain than white patients in the INSIGHT and higher excess burdens of pressure ulcers in the OneFlorida+.

**Baseline pre-existing conditions.** Overall, we observed a higher excess burden (quantified by the difference of cumulative incidences) in patients with any baseline pre-existing conditions in Elixhauser comorbidity groups (See Method) than in patients without any assessed comorbidities or PASC-like symptoms (denoted as healthy patients). There were also varying excess burdens of different potential PASC conditions associated with patients with different pre-existing conditions. For example, patients with coronary artery disease (CAD), chronic kidney disease (CKD), or chronic pulmonary disease (CPD) had higher burdens of pulmonary fibrosis, malaise, and fatigue. Even for healthy patients without documented baseline Elixhauser comorbidities, we observed incident dyspnea, pulmonary fibrosis, and chest pain burdens in both two cohorts, and diabetes burden in the INSIGHT cohort.

**Different waves.** We further stratified the excess burden of PASCs over different waves associated with different SARS-CoV-2 variants. As shown in Supplementary Fig. 1, we defined three waves March 1, 2020, to September 30, 2020, October 1, 2020, to May 31, 2021, and June 1, 2021, to November 30, 2021. The most common SARS-CoV-2 genotypes prevalent in the 1st and 3rd waves were the ancestral strain and the Delta variant respectively, and the 2nd was a mixture of the Alpha variant and others, according to the CDC[18]. We summarized the results in Supplementary Fig. 6. We found the first two waves contributed to the most significant PASC burdens in the INSIGHT and all three waves for the OneFlorida+. The dyspnea condition was consistently significant over three waves and across the two cohorts.

## Negative controls

We employed negative outcome controls[19,20] in both the INSIGHT and OneFlorida+ cohorts to rule out potential residual confoundings. We examined the adjusted risk of a range of clinical outcomes (e.g., injury

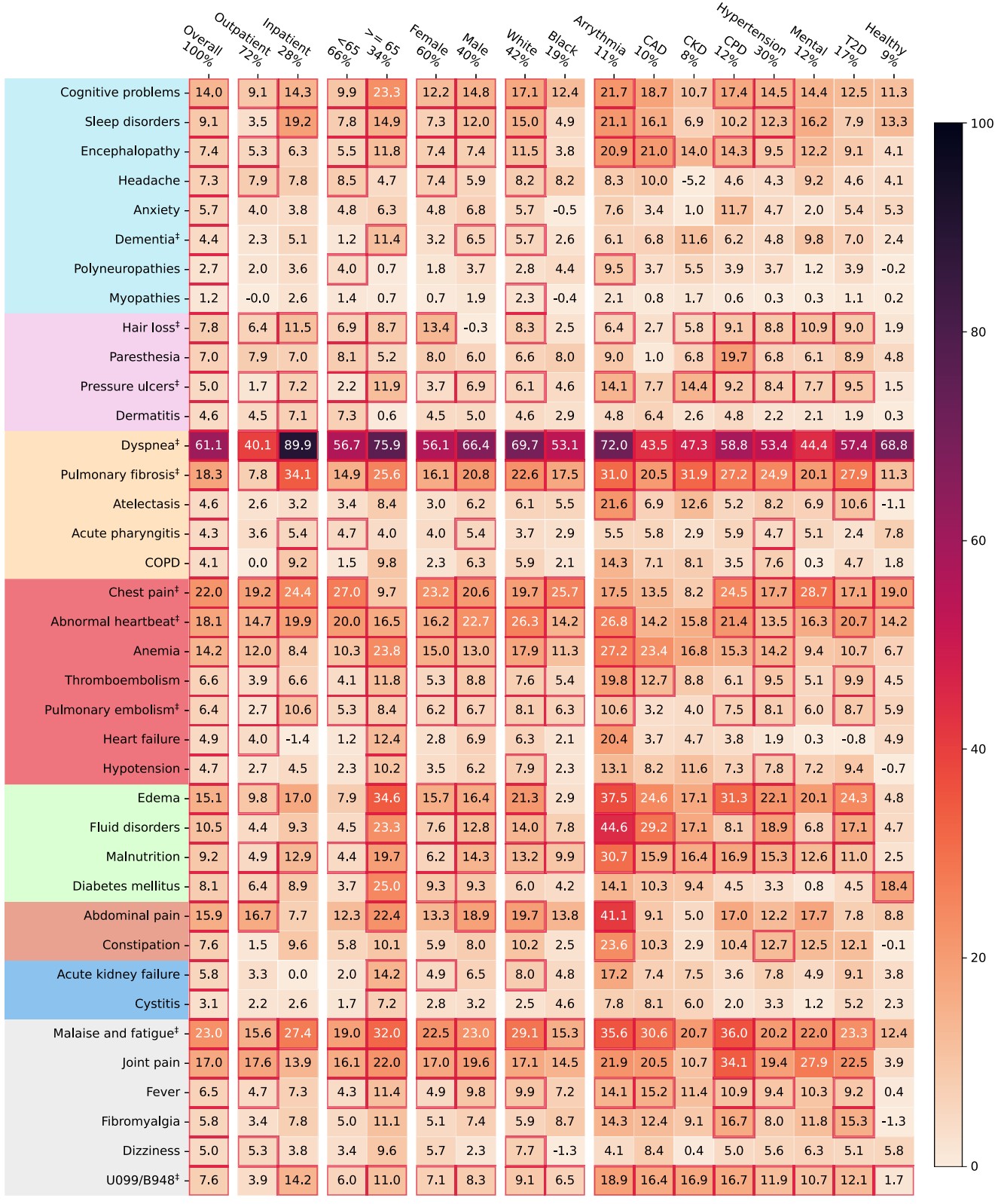

due to external causes and neoplasms-related outcomes) where no association was expected with SARS-CoV-2 infection based on existing knowledge. We followed the same procedure as in screening potential PASC conditions and estimated the adjusted risk in both exposure groups. We found no significant association between any of the negative outcomes and SARS-CoV-2 infection after the acute phase as shown in Supplementary Table 1.

## Sensitivity analysis

To test the robustness of our results, we conducted a series of sensitivity analyses. Firstly, we investigated how the identified PASC conditions will change when using different screening criteria in the context of multiple comparisons. Here, we considered the false discovery rate-based method (threshold was 0.05)−Benjamini−Yekutieli method (BY)[21], which is less strict than the Bonferroni correction

**Fig. 4 | Stratified analysis of adjusted excess burden of post-acute sequelae of SARS-CoV-2 infection (PASC) over different subgroups, the INSIGHT cohort, from March 2020 to November 2021.** The adjusted excess burden is measured by the difference in the adjusted cumulative incidence per 1000 between two exposure subgroups. Subgroups were stratified by their acute severity status, age groups, gender, race groups, and baseline pre-existing conditions. Different color panels represent different organ systems, including (from top to bottom): the nervous system, skin, respiratory system, circulatory system, blood-forming organs, endocrine and metabolic, digestive system, genitourinary system, and general signs. CAD coronary artery disease, CKD chronic kidney disease, CPD chronic pulmonary disease, T2D diabetes type 2, Healthy: no documented pre-existing conditions and no PASC-like symptoms at baseline. Two ICD-10 diagnosis codes B948 (sequelae of other specified infectious and parasitic diseases) and U099 (post-COVID-19 condition, unspecified) were also used to compare general post-acute sequelae of SARS-CoV-2 infection in different groups. The conditions with their aHRs' *P*-value < $8.39 \times 10^{-5}$ (the Bonferroni-corrected significance threshold) were highlighted in red squares. The PASC conditions also identified in OneFlorida+ were marked by ‡ symbols. The fraction of the subgroup population was shown at the top.

approach we used in our main analysis. By using the BY method, in addition to our identified PASC conditions reported in the main text, we found other seven potentially significant PASC diagnoses (cerebral ischemia, acute myocardial infarction, pulmonary heart disease, myocarditis, nausea, Gastro-esophageal reflux disease, genitourinary problems) from the INSIGHT and one (acute pharyngitis) PASC diagnosis from the OneFlorida + . We further investigated how sensitive the final PASC conditions are if we lift the constraints of at least 100 identified cases in the SARS-CoV-2 group. We only found one additional condition "foot drop" from INSIGHT. We summarized the additional PASC conditions identified in these sensitivity analyses in Supplementary Fig. 7 and Supplementary Data 4–5. Overall, only acute pharyngitis was replicated across two cohorts, and these additional conditions showed higher risk in the INSIGHT than in the OneFlorida+. In addition, if using less conservative BY correction, we still cannot find any statistically significant negative signals (adjusted hazard ratio <1) from our initial screen list on the INSIGHT cohort (Supplementary Data 4).

We also investigated how the PASC risks in two cohorts will change if we change our baseline covariates modeling. First, we adjusted for additional baseline covariates by capturing the baseline SARS-CoV-2 vaccination status. We categorized baseline vaccination status into fully vaccinated, partially vaccinated, and no evidence of vaccination (See the population characteristics and the covariates definition in Supplementary Table 4). We replicated our primary analyses on two cohorts by adjusting for these additional covariates. As shown in Supplementary Fig. 8, adjusting for these additional baseline vaccination covariates had little impact on the adjusted hazard ratios, largely because nearly half of the study patients got infected before the vaccine was available (early December 2020) and more than 90% of populations had no evidence of vaccination recorded in the EHR systems (Supplementary Table 4). Second, instead of categorizing the index day into different periods in our primary analysis, we further adjusted for index day as a continuous variable using the cubic B-spline model[22,23]. We compared aHR using different modeling of index day (categorization vs. cubic B-spline) in Supplementary Fig. 9, and we found similar aHRs and got the same set of PASC conditions identified from both datasets as the primary analysis (Supplementary Fig. 9, conditions marked by ‡ symbols). Third, we compared nonlinear PS modeling as a sensitivity analysis and linear PS modeling in our primary analysis. Specifically, we adopted the gradient boosting decision trees for modeling PS (Method Section)[24], and we reported aHR in Supplementary Fig. 10. Again, consistent results in terms of aHR and replicated PASC conditions were observed.

Lastly, different PASC risk patterns might derive from sample size differences between the two cohorts. To rule out this, we downsampled the INSIGHT cohorts to have the sample number of patients as the OneFlorida+ and replicated our primary analyses on the downsampled cohorts. Specifically, we downsampled the SARS-CoV-2 positive group in the INSIGHT cohort from 35,275 to 22,341, and the negative group from 326,126 to 177,010, leading to the same number of patients as in the OneFlorida+ cohort. We compared the aHR when using the downsampled cohorts versus using the original cohort in Supplementary Fig. 11. Again, we observed the same PASC risk patterns

and got the same set of replicated PASC conditions as the primary analysis (Supplementary Fig. 11, conditions marked by ‡ symbols).

## Discussion

In this study, we developed a data-driven approach to identify a broad spectrum of clinical abnormalities (incident diagnoses and medication use) experienced by SARS-CoV-2 infected patients who survived beyond the first 30 days of the infection. The clinical EHRs from two large PCORnet clinical research networks, INSIGHT, and OneFlorida+, were leveraged in our study to investigate the heterogeneity of potential PASC conditions over different patient populations. This differentiates our study from prior studies that focused on a specific patient population (e.g., Al-Aly et al.[7]. focused on the US veteran population with 87.91% males). There are also several studies focusing on a more discrete set of potential PASC conditions such as mental health problems[9,25], cardiovascular problems[8], diabetes[26], and kidney problems[27] among others. All these studies investigated a single dataset.

With a screening pipeline based on high-throughput trial emulations and stabilized IPTW weights-based adjustment, we identified a broad spectrum of diagnoses and medication use that exhibited higher adjusted hazard ratios and excess burdens in SARS-CoV-2 infected patients in the post-acute period compared to non-infected patients. These diagnoses and medications spanned a wide range of organ systems (Fig. 2), suggesting that PASC is a multi-organ disease. Diagnoses with high adjusted hazard ratios included respiratory problems (e.g., dyspnea, pulmonary fibrosis, atelectasis, COPD), dermatologic problems (e.g., hair loss, paresthesia, pressure ulcers, and dermatitis), cardiovascular problems (e.g., pulmonary embolism, thromboembolism, chest pain, abnormal heartbeat, and heart failure), nervous system problems (e.g., encephalopathy, dementia, sleep disorders, encephalopathy, cognitive problems, polyneuropathies, myopathies, and anxiety), and general symptoms (e.g., malaise, fatigue, fever, fibromyalgia, dizziness, joint pain, and U099/B948 codes). In addition to diagnoses, we also observed increased incident prescription risk in a diverse set of medications, including asthma drugs (e.g., vilanterol trifenatate and fluticasone furoate), cough drugs (e.g., dextromethorphan and benzonatate), anticoagulants (e.g., apixaban, heparin, and aspirin), diabetic drugs (e.g., insulin and metformin), drugs for constipation (e.g., magnesium hydroxide), drugs for vomiting (e.g., trimethobenzamide), pain medications (e.g., menthol, ibuprofen, and acetaminophen), drugs for treating skin problems (e.g., witch hazel and collagenase), and insomnia drugs (e.g., melatonin). These conditions and medications showed a higher incidence of diagnosis or use after infection than the non-infected control group, suggesting that these could be likely post-acute sequelae of SARS-CoV-2 infection (PASC) conditions.

We have also performed detailed stratified analyses on the adjusted excess burden of different potential PASC diagnoses over different groups defined by age, sex, race, acute severity of SARS-CoV-2 infection, baseline comorbidity conditions, and temporal waves considering different SARS-CoV-2 variants of concerns. Our results showed that, in both the INSIGHT and OneFlorida+ cohorts, hospitalized patients demonstrated more excess cases of potential PASC

diagnoses and medications (compared to non-infected controls) than non-hospitalized patients, especially for respiratory conditions. Older patients also had higher excess cases of PASC conditions than younger patients, as did female and non-white patients. These observations were consistent with prior studies[7,17]. Patients with co-morbidities had a higher incidence and number of putative post-acute SARS-CoV-2 conditions. Furthermore, the distribution of post-acute conditions varied across distinct co-morbidities. We observed that dyspnea consistently showed the highest excess burden across all patients regardless of co-morbidity status. Patients with baseline cardiac problems (arrhythmia and coronary heart disease), diabetes, and chronic kidney disease (CKD) demonstrated higher burdens of a more diverse set of potential PASC conditions than other comorbidity groups. Patients without pre-existing conditions at baseline had higher dyspnea, chest pain, diabetes, malaise, and fatigue burdens compared with control patients.

We observed heterogeneity after replicating the same analysis to the OneFlorida+ cohort. Overall, aHRs on all potential PASC diagnoses identified from the INSIGHT cohort were higher than in the OneFlorida+ cohort. In particular, rates of pulmonary fibrosis and thromboembolism were 50% higher in the INSIGHT cohort than in the OneFlorida+ cohort. In addition, 38 diagnoses and 59 medications were identified as potential PASCs in the INSIGHT cohort compared to the 11 diagnoses and 9 medications identified in the OneFlorida+ cohort. The conditions replicated on both datasets were dementia, hair loss, pressure ulcers, pulmonary fibrosis, dyspnea, pulmonary embolism, chest pain, abnormal heartbeat, malaise, and fatigue. Potential reasons accounting for this heterogeneity of PASC conditions include distinct patient characteristics and different periods of infections which could have led to differential use of therapeutics and vaccination that could alter the trajectory of PASC. The SARS-CoV-2 infected patients in the OneFlorida+ cohort were younger (median age 50 (34–64)) than those in the INSIGHT cohort (median age 55 (38–68)). Younger adults are at lower risk for PASC than older adults. Patients in the OneFlorida+ cohort were also much more socially disadvantaged on average. Their median ADI ranking value was almost four times higher than the median ADI of patients in the INSIGHT cohort. Disadvantaged social conditions can be associated with delayed or no care access and initiation of treatment for PASC conditions; therefore, OneFlorida+ patients might be less likely to present for care during a relatively short post-acute phase, leading to an undercount of potential PASC conditions and medication use in that population. Disadvantaged social conditions can also be associated with poorer baseline conditions even in non-infected patients, leading to nonsignificant excess burden between infected and non-infected groups in the OneFlorida+ cohort.

The treatment standard for COVID-19 evolved over time[28,29]. For example, there was a demonstrable higher use of corticosteroids in the Florida cohort compared with the NYC cohort. Patients who received timely and appropriate treatment for COVID-19 in the acute phase could be less likely to develop PASC in the post-acute phase. Early evidence showed that vaccinations for COVID-19 significantly reduced the likelihood of getting PASC conditions[30]. It should be noted that NYC had a higher incident burden of SARS-CoV-2 infection prior to the widespread availability of vaccinations in December 2020. In comparison, OneFlorida+ had a high burden of incident SARS-CoV-2 infections after December 2020.

Our study has several strengths. First, this study examined PASC in a large population of general adult patients using a data-driven approach to identify a broad list of potential PASC. Second, this study incorporated EHR data from two large-scale clinical research networks covering patients from distinct geographic regions in the US with very different characteristics, allowing us to highlight the heterogeneity of PASC manifestations in terms of diagnoses and medications over two different populations

thereby improving generalizability. Third, from March 2020 to November 2021 (the enrollment period of our study), the US went through COVID-19 waves associated with different SARS-CoV-2 virus variants demonstrating different epidemiological and clinical characteristics. Our INSIGHT and OneFlorida+ cohorts contained robust patient populations in New York and Florida, representing the different waves of SARS-CoV-2 infected cases in the US. This temporal difference is another important factor accounting for the different observations from the two cohorts, in addition to their different demographic and geographic characteristics.

Existing studies with comparable sample sizes are the study from Al-Aly et al.[7,17], which focused on the veteran (VA) population with mostly males (90.5%), and Cohen et al.[31]. focusing on older patients (age ≥ 65) enrolled in the Medicare Advantage plan (administrative claims). Compared with the VA study, we found several new conditions including dementia, pulmonary fibrosis, and pulmonary embolism, which were not reported in the VA study. In addition, we also found differences in terms of adjusted excess burden per 1000 patients between our studies and the VA's, e.g., hair loss (7.8 vs. 5.2 vs. 0.2 (INSIGHT vs. OneFlorida+ vs. VA, adjusted excess burden per 1000)), dyspnea (61.1 vs. 41.2 vs. 28.8), chest pain (22.0 vs. 15.8 vs. 13.8), abnormal heartbeat (18.1 vs. 10.2 vs. 7.85), and malaise and fatigue (23 vs. 12.5 vs. 13.6). We further compared some replicated conditions in both our study and Cohen's Medicare study based on the subpopulation with age greater than or equal to 65, including dementia (11,4 vs. 15.4 vs. 13.6 (INSIGHT vs. OneFlorida+ vs. Medicare, an excess burden per 1000, age ≥ 65 subgroup)), pulmonary embolism (8.4 vs. 7.8 vs. 9.4), abnormal heartbeat (16.5 vs. 5.5 vs. 16.4), malaise and fatigue (32.0 vs. 17.2 vs. 44.5). In addition, our study followed patients till November 2021, longer than the VA study (till May 2021)[7,17] or the Cohen's study (till December 2020)[31].

There are also several limitations. First, our study was based on observational data analysis, and patients' assignments to particular exposure groups were not randomized. Although we have tried to balance high-dimensional hypothetical confounders and obtained consistent results from several negative outcome control analyses across two datasets, there is still a possibility that chance finding could exist and it is challenging to draw causal conclusions from observational data analysis. Hopefully, the conclusions from our study can serve as effective hypotheses to trigger future biological mechanistic studies. Second, our study included the patient population from the NYC and Florida areas, which may not be representative of other geographical regions of the US or other countries. Third, the PASC is currently defined in the RECOVER protocols as "ongoing, relapsing, or new symptoms, or other health effects occurring after the acute phase of SARS-CoV-2 infection" (https://recovercovid.org/). Our study only studied incident events, and the worsening and relapsing conditions were left for future investigations. Fourth, the way these CCSR categories were defined may not reflect the actual co-occurring risk of the individual conditions contained in each in the context of PASC. In addition, our study period was from March 2020 to November 2021, which did not include patients infected during the phase dominated by the Omicron variants of SARS-CoV-2. Lastly, our main analyses did not include information on vaccination status in our primary analyses and we left it as our future focus.

In conclusion, this study demonstrated that adult patients surviving beyond 30 days of their SARS-CoV-2 infection exhibited high incident risks and burdens across a broad range of conditions and signs. Our findings verified that PASC is a complex condition involving multiple organ systems. There was surprising geographic heterogeneity of PASC as well as patient sub-group heterogeneity. This study provides additional insights into our understanding of PASC and highlights the need for further research to support the diagnosis,

prevention, and treatment of the post-acute sequelae of SARS-CoV-2 infection.

## Methods

### Data

This study used two large-scale de-identified real-world EHR datasets (RWDs) from the INSIGHT Clinical Research Network (CRN)[13] and the OneFlorida+ CRN[14]. The INSIGHT CRN contained longitudinal clinical data of approximately 12 million patients in the New York City metropolitan area, and the OneFlorida+ CRN contained the EHR data of nearly 15 million patients from Florida and selected cities in Georgia and Alabama. The use of the INSIGHT data was approved by the Institutional Review Board (IRB) of Weill Cornell Medicine following NIH protocol 21-10-95-380 with protocol title: Adult PCORnet-PASC Response to the Proposed Revised Milestones for the PASC EHR/ORWD Teams (RECOVER). The use of the OneFlorida+ data for this study was approved under the University of Florida IRB number IRB202001831. All EHR used in this study were appropriately dei-dentified and thus no informed consent from patients was obtained.

### High-throughput screening for potential post-acute sequelae of SARS-CoV-2 (PASC)

To systematically identify potential PASCs, we examined a total of 596 incident diagnoses (Supplemental Data 2) and medication use (Supplemental Data 4) in the SARS-CoV-2 infected patients from 31 days to 180 days after their acute infection. For each incident diagnostic category or medication use, we constructed an outcome-specific cohort including both SARS-CoV-2 infected patients and non-infected patients who did not have the corresponding diagnostic category or medication use at baseline and assessed its incident risk in the post-acute phase (see Fig. 1 for a graphical illustration of our pipeline). Thus, leveraging RWDs, we evaluated the impact of SARS-CoV-2 infection (as exposures) in the post-acute period using each target outcome, leading to 596 independent analyses, aiming to generate PASC hypotheses in a high-throughput and data-driven manner. The key components of our high-throughput screening framework are summarized in Supplementary Table 2 and detailed as follows.

**Eligibility criteria and exposure strategies.** We included patients with at least one SARS-CoV-2 polymerase-chain-reaction (PCR) or antigen laboratory test between March 01, 2020, and November 30, 2021, for both cohorts. Other eligibility criteria included an age of at least 20 years old, at least one diagnosis code within three years to seven days before the index date (referred to as the baseline period), and at least one diagnosis code from 31 days to 180 days after the index date (referred to as the post-acute phase or follow-up period), to ensure that patients were connected to the healthcare system and were being observed during the study period. Two exposure groups were the SARS-CoV-2 infected group and the non-infected group. The SARS-CoV-2 infected group included patients with a positive SARS-CoV-2 PCR or antigen laboratory test. The index date of the infected group was defined as the date of the first documented positive PCR or antigen test. The non-infected group included patients whose SARS-CoV-2 PCR or Antigen tests were all negative throughout the entire study period with no documented COVID-19-related diagnoses. The index date for patients in the non-infected group was defined as the date of the first negative PCR or antigen test. See Supplementary Data 1 for the list of COVID-19-related LOINC laboratory codes and ICD-10 diagnosis codes used for cohort selection.

**Group assignment and baseline covariates.** Patients in the two exposure groups, namely the SARS-Cov-2 infected group and the non-infected group, were assumed exchangeable after adjusting for high-dimensional baseline covariates as hypothetical confounders. The collected baseline covariates included age (categorized into 20–39 years, 40–54 years, 55–4 years, 65–74 years, 75–84 years, 85 years and older), gender (female, male, other/missing), race (Asian, Black or African American, White, other, missing), ethnicity (Hispanic, not Hispanic, other/missing). The race and ethnicity were self-reported. The national-level area deprivation index (ADI) was used to capture the socioeconomic disadvantage of patients' residential neighborhoods[15]. We used a 9-digit zip code to link to the national ADI percentiles (ranked from 1 to 100). We imputed the missing ADI value with the median ADI per site. The ADI ranks from 1 to 100, with 1 and 100 indicating the lowest and highest level of disadvantage[15]. Health-care utilization was measured as the number of inpatients, outpatient, and emergency encounters (0 visits, 1 or 2 visits, 3 or 4 visits, and 5 or more visits for each encounter type) respectively. In addition, periods (March 2020–June 2020, July 2020–October 2020, November 2020–February 2021, March 2021–June 2021, July 2021–November 2021) of the index date were used to account for potentially different stages of the pandemic. The body mass index (BMI) was categorized into underweight ($<18.5\,\mathrm{kg/m^2}$), normal weight ($18.5\,\mathrm{kg/m^2}$–$24.9\,\mathrm{kg/m^2}$), overweight ($25.0\,\mathrm{kg/m^2}$–$29.9\,\mathrm{kg/m^2}$), obesity ($\geq 30.0\,\mathrm{kg/m^2}$), and missing according to the CDC guideline for adults[32].

We also collected a wide range of baseline comorbidities based on a tailored list of the Elixhauser comorbidities[16] and related drug categories, including alcohol abuse, anemia, arrhythmia, asthma, cancer, chronic kidney disease, chronic pulmonary disorders, cirrhosis, coagulopathy, congestive heart failure, chronic obstructive pulmonary disease, coronary artery disease, dementia, diabetes type 1, diabetes type 2, end-stage renal disease on dialysis, hemiplegia, HIV, hypertension, hypertension and type 1 or 2 diabetes diagnosis, inflammatory bowel disorder, lupus or systemic lupus erythematosus, mental health disorders, multiple sclerosis, Parkinson's disease, peripheral vascular disorders, pregnant, pulmonary circulation disorder, rheumatoid arthritis, seizure/epilepsy, severe obesity (BMI $>= 40\,\mathrm{kg/m^2}$), weight loss, Down's syndrome, other substance abuse, cystic fibrosis, autism, sickle cell, corticosteroid drug prescriptions, immunosuppressant drug prescriptions. Patients were defined as having a condition if they had at least two corresponding diagnoses documented in the three years before the index event.

**Follow-up period.** We followed each patient from 31 days after his/her index date until the day of the first target outcome, documented death, loss of follow-up in the database, 180 days after the baseline, or the end of our observational window (December 31, 2021), whichever came first.

**Diagnosis categories for screening potential PASC conditions.** We examined an initial list of potential adult PASC diagnostic outcomes for screening, which contained 137 diagnostic categories. A team of clinicians built our initial screening list based on the Clinical Classifications Software Refined (CCSR) v2022.1 covering all the 66,534 ICD-10-CM Diagnoses, and removed codes that cannot be attributed to COVID-19 (e.g., HIV, tuberculosis, infection by non-COVID causes, neoplasms, injury due to external causes), and systematically added parent codes (e.g., the first 3-digits of ICD-10 codes) of potential PASC diagnosis codes. The full list of our investigative diagnosis codes is provided in Supplementary Data 2 and includes 6466 codes.

**Medications for screening potential PASC conditions.** We examined an initial list of potential adult PASC medication outcomes for screening, which contained 459 drug categories classified by their active ingredients. We collected real-world drug prescription data from our EHR datasets, mapped drugs into their active ingredients, and selected drug ingredients prescribed for at least 100 patients in the SARS-CoV-2 positive group, which led to 434 active drug ingredients. We further considered another 25 categories of medications used during treatment for COVID-19, including anti-platelet therapy,

aspirin, colchicine, corticosteroids, dexamethasone, and heparin, which were potentially identified by both prescription records and procedure records.

**Contrasts of PASC outcomes.** Adjusted hazard ratio with 95% confidence intervals and excess burden for each incident PASC diagnosis or medication were calculated at the end of the follow-up period. Adjusted cumulative incidence for each exposure group was also reported.

**High-throughput screening pipeline for PASC.** We systematically examined the 137 diagnosis categories and 459 medication ingredients using our pipeline shown in Fig. 1.

**Statistical analyses for high-throughput hypotheses generation**
**Inverse probability of treatment weighting for adjustment.** We built a propensity score (PS) model—the probability of assignment of a particular exposure group conditioned on baseline covariates—for each target outcome. Based on the estimated PS values, we then used stabilized inverse probability of treatment weighting (IPTW)[33] to re-weight patients in exposure and control groups, aiming to balance the two groups on baseline covariates after re-weighting. If we use $\mathbf{X}$, $Z$ to represent the observed baseline covariates and the assignment of exposure ($Z = 1$) and control groups ($Z = 0$), the PS is defined as $P_\theta(Z = 1|\mathbf{X})$ and the stabilized IPTW is shown in Eq. (1).

$$w = \frac{Z^*P(Z = 1)}{P_\theta(Z = 1|\mathbf{X})} + \frac{(1 - Z)^*P(Z = 0)}{1 - P_\theta(Z = 1|\mathbf{X})} \qquad (1)$$

To deal with potentially large weight and thus large variability of estimated effects, we adopted the stabilized IPTW, which shrinks the conventional IPTW by a smaller-than-1 factor $P(Z)$. We further trimmed extreme weights beyond their 1st or 99th percentiles to control for potentially large weights to reduce variability[34]. We used standardized mean difference (SMD) to quantify the goodness-of-balance of covariates over two groups as shown in Eq. (2) and used SMD < 0.1 as the threshold for balancing diagnostics. The SMD was calculated before and after IPTW re-weighting, and the results are provided in Supplementary Data 4 and 5.

$$\text{SMD}(\mathbf{X}_1, \mathbf{X}_0) = \frac{E[\mathbf{X}_1] - E[\mathbf{X}_0]}{\sqrt[2]{(\text{Var}(\mathbf{X}_1) + \text{Var}(\mathbf{X}_0))/2}} \qquad (2)$$

We used logistic regression with the L2 penalty term for PS calculation, with the optimal regularization strength determined through grid search over hyper-parameter space ($10^{-2}$, $10^{-1.5}$, $10^{-1}$, $10^{-0.5}$, 1, $10^{0.5}$, $10^{1}$, $10^{1.5}$, $10^{2}$, and no penalty). We have shown in our previous study that a better PS model can be selected by considering both the goodness-of-balance performance and the goodness-of-fit performance[35]. Here, we show details of the cross-validation pipeline for PS model training, selection, and validation in Supplementary Table 3, which can achieve better goodness-of-balance performance compared with the PS model selected by other machine learning model selection strategies in trial emulations[35].

**Statistical analysis for outcome contrasts.** The adjusted hazard ratio (aHR) was estimated by Cox proportional hazard model with the abovementioned stabilized IPTW weights with trimming 1st and 99th extreme values. The cumulative incidence was estimated by the Aalen-Johansen model[36] considering death to be a competing risk for the target outcomes, adjusted by the same IPTW weights as used in aHR. The excess burden was defined as the difference in adjusted cumulative incidences in different exposure groups.

**Screening criteria for likely PASC conditions.** To reduce the chance of false positive discovery, we adhered to the following screening criteria: Only diagnoses and medications with (1) adjusted hazard ratios larger than 1, (2) *P*-value $< 8.39 \times 10^{-5}$ (significance level corrected by Bonferroni method, namely $\frac{0.05}{596}$, for multiple comparisons) will be retained as likely PASC conditions. Further, we required a minimum number of PASC cases that appeared at least 100 times in the post-acute period of the SARS-CoV-2 infected group in each emulated trial.

**Sensitivity analyses.** First, we investigated to what extent the identified PASC conditions will change by using different correction methods for the multiple comparisons problems. The Bonferroni method (BF) controls the familywise error rate (FWER), namely the probability of making one or more false discoveries, which is a very stringent method, leading to fewer discoveries. We also considered the false discovery rate-based method (FDR)−Benjamini−Yekutieli method (BY)[21]−which is less stringent than the FWER-based method and requires no assumptions about the correlations of different tests. We also checked how the results will change by further lifting the criterion which required at least 100 cases in the SARS-CoV-2 infected group. In addition, we also check significant conditions with aHR <1. Second, we considered different baseline covariates modeling, including (a) adjusting for additional baseline vaccination status, (b) modeling index day by the cubic B-spline method using 7 spline basis functions of polynomial order 3[22,23], and (c) modeling propensity score by using (non-linear) gradient boosting decision trees[24]. Third, we investigated the potential impact of sample size difference by replicating the analyses on a downsampled INSIGHT cohort, which had the same number of patients as in the OneFlorida+. The best gradient-boosting decision tree model was selected from a set of models defined by hyper-parameters including maximum depth (3, 4, 5), learning rate (0.01, 0.26, 0.51, 0.76), max number of leaves in one tree (5, 20, 35), and the minimal number of samples in one leaf (100, 200, 300), using the cross-validation algorithm as detailed in Supplementary Table 3.

**Subgroup analysis and negative outcome controls.** The subgroup analysis was conducted by stratifying patients (in both SARS-CoV-2 infected and non-infected groups) by their age, race, gender, the severity of acute infection (outpatient or inpatient), and pre-existing conditions. We included a population with no documented pre-existing conditions or PASC-like symptoms at baseline, denoted as healthy. We also stratified patients by different waves to check the potential heterogeneity in different variants of concerns. To explore the possible existence of residual confounding, we estimated the adjusted hazard ratio of non-PASC outcomes following negative outcome control framework[19,20].

**Reporting summary**
Further information on research design is available in the Nature Portfolio Reporting Summary linked to this article.

## Data availability

The INSIGHT data can be requested through https://insightcrn.org/. The OneFlorida+ data can be requested through https://onefloridaconsortium.org. Both the INSIGHT and the OneFlorida+ data are HIPAA-limited. Therefore, data use agreements must be established with the INSIGHT and OneFlorida+ networks.

## Code availability

For reproducibility, our codes are available at https://github.com/calvin-zcx/pasc_phenotype[37]. We used Python 3.9, python package lifelines-0.2666 for survival analysis, and scikit-learn-0.2318 for machine learning models.

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

## Acknowledgements

This research was funded by the National Institutes of Health (NIH) Agreement OTA. OT2HL161847 (contract number EHR-01-21) as part of the Researching COVID to Enhance. Recovery (RECOVER) research program. The PCORnet® Study reported in this work was conducted using PCORnet®, the National Patient-Centered Clinical Research Network. PCORnet® has been developed with funding from the Patient-Centered Outcomes Research Institute® (PCORI®). This work was conducted through the use of data from the INSIGHT Clinical Research Network and supported in part by the Patient-Centered Outcomes Research Institute (PCORI) PCORnet grant to the INSIGHT Clinical Research Network (Grant # RI-CORNELL-01-MC). The statements presented in this work are solely the responsibility of the author(s) and do not necessarily represent the views of other organizations participating in, collaborating with, or funding PCORnet® or of the Patient-Centered Outcomes Research Institute® (PCORI®).

## Author contributions

C.Z. and F.W. proposed the initial idea. C.Z. designed and implemented the framework and analyzed the results. D.M. and MGW set up the data infrastructure and analytics environment for INSIGHT. J.B. set up the data infrastructure and analytics environment for OneFlorida + .

C.Z. and J.X. preprocessed the INSIGHT and OneFlorida+ data. Y.Z. helped with the statistical analysis. E.J.S., D.K., A.S.N., E.A.S., R.L.R., J.P.B., K.L., M.G.W., T.W.C., and R.K. provided clinical inputs on data, study design, and results interpretation. C.Z. drafted the initial manuscript. F.W., T.W.C., and R.K. made critical revisions. All authors have provided feedback and proofread the final version of the paper.

## Competing interests

The authors declare no competing interests.
