## [Peer Review File · Nature Communications]

Data-driven analysis to understand Long COVID using electronic health records from the RECOVER initiativeEditorial Note: Parts of this Peer Review File have been redacted as indicated to remove third-party material where no permission to publish could be obtained.

REVIEWER COMMENTS

Reviewer #1 (Remarks to the Author):

This is an interesting real world electronic health record study of events recorded in the health care records of two very large datasets in New York and Florida between 31 and 180 days after COVID-19 compared to individuals who never tested positive over the study period.

Advantages are the study size and the attempt to use inverse probability score weighting to account for clear evident differences between groups.

None of the findings are particularly surprising in the context of the international literature published so far, with the exception of the fact that the rates overall of pasc are really quite low – even dyspnoea is reported in only 8% of hospitalised cases after 180 days. The fact that access issues may undermine the reporting in the health record is important, making interpretation difficult without a benchmark using population-based data. Indeed, it is disappointing to note that there is no reference to other large scale electronic health record datasets or population based samples, which are already published. I'm interested that in the conclusion the authors states these data had exhibited "high incident risks". I think their overall incident data presented are relatively low compared to other datasets, unless I have misunderstood the data presented.

Overall, while not exactly novel in terms of findings, this is an interesting contribution which validates existing publications – in particular, increased risk of thromboembolism and lung fibrosis, but also many non-specific conditions. I was interested that they did not report increased incidence of for example new myocardial infarction or new diabetes diagnosis, which has been found in other datasets. Indeed, it was disappointing to see only positive results reported. Were there any interesting exceptions here from existing literature?

In the population statistics section, would be useful to know how representative of the general population people registered with these patient centred clinical research networks are. Do they just include people with access to health insurance?

What are the statistics in the results section - ?OR ?HR – it took me til the methods to get what you'd done, due to the English in the description. Eg "We reported their incident risks in the adjusted hazard ratio (Fig. 2) and excess burdens in the adjusted excess cumulative incidence" should probably be "Incident risks are reported as adjusted hazard ratios (Fig. 2) and adjusted excess cumulative incidences over 180 days."

On this point about excess cumulative incidences : further amendments to English are needed in the paragraphs discussing these subgroup analyses. I am also not entirely sure that you are presenting the "excess cumulative incidence" in Figure 3, or the cumulative incidences by group. You state "We estimated the adjusted cumulative incidence of each potential PASC diagnosis per 1,000 patients at 180 days in different groups and compared the excess burden of it18, which is the difference between the adjusted cumulative incidences of a specific diagnosis in the SARS-CoV-2 infected subpopulation and the corresponding control subgroup" What therefore is provided in subjects overall? It looks like from many of the figures reported are cumulative incidence rather than the difference, or excess, as the comparative subgroups often straddle the overall, (but this is not uniformly the case, eg cognition and dementia). It's also really important that you label this as per 1000 patients, in the legend, and preferably by the scale, which seems to imply a 100 top value, which is misleading in my view.

The causal inference procedure accounted for the population differences between those who tested positive and those who did not, using inverse propensity score weighting, (at least for the Hazard ratios, but I am not certain about the cumulative incidences). Therefore findings should not be influenced by differences in populations testing positive (which may be affected by testing availability and exposure). However, these differences are quite marked, and did make me think throughout that I would prefer not to have seen the words "causal inference" used so shamelessly. Ultimately this is a large electronic health data study with some clever stats, but its very clearly observational and I would not make causal inferences from it.

This is further underlined by the fact that the two cohorts had quite differing baseline characteristics between the positive and negative groups, which it appears may then play out in the results.

Reading through I found it difficult to work out the differences in timings of infection between the two cohorts, and the supplementary figure was helpful and confirmatory. Table one points out the

time periods when data was collected and there are interesting differences in the time periods for the majority of positive cases between the two cohorts. This then may be reflected in the descriptive data and the differences in subsequent findings in the cohorts. This as well as the other cohort differences alluded to above make interpretation difficult and the study purely descriptive. The authors do attempt to discuss this in relation to differences between variants and in the advent of vaccination. However, with such a large dataset it is a shame not to segment into variant / vaccination groupings.

For the international reader some context on the ADI would be helpful. What do the differences between the two cohorts mean (need to know what the range and interpretation is).

Reviewer #2 (Remarks to the Author):

The authors examine the association of COVID-19 with 137 diagnoses and 459 medications (I think there is a typo when they say 359 elsewhere in the paper) in the post-acute period, which they define as from 30 to 180 days after COVID-19 diagnosis. They refer to these collectively as post-acute COVID-19 sequelae (PASCs). Their analyses are based on a "high-throughput causal inference pipeline", which is implemented in two electronic health record (EHR) datasets. Only diagnoses and medications with (1) adjusted hazard ratios larger than 1, (2) P-value smaller than 3.6×10^{-4} for diagnoses and 1.4×10^{-4} (for medications) (the authors made a Bonferroni method for multiple testing) were retained as potential PASCs. The reported results are mainly based on the larger INSIGHT cohort (~35k SARS-CoV-2 positive, ~326k negative). The authors report that results are rather different in the smaller OneFlorida+ cohort (~22k positive, ~177k negative).

Major comments.

1. The wide range of post-COVID events analysed is a strength of the paper. However, it means that in the results section there is a strong emphasis on the minority of associations that were identified as PASCs. We have no information on the specificity of the identified associations, because the non-selected results are, as not reported in detail. The authors should include information about the diagnoses and medications that were not identified as PASCs. They should also comment on the role of the number of events, because if there is a small number of events the association has to be very large to achieve a p value smaller than the threshold. It is likely that many potential PASCs with large HRs were not considered positive. The Bonferroni correction is based on an assumption that the joint null hypothesis is true – this is not plausible for post-COVID events.
2. The same issue applies to subgroup analyses. Even if associations were identical within subgroups, with the same hazard ratio as overall, they could be designated as not associated because the significance within subgroups no longer reaches the threshold.
3. The authors have implemented a "causal inference" pipeline in which they aim to emulate a target trial. However the flow chart in Figure 1 implies that there are two stages of selection before patients enter the target trial: first the restriction to those who tested, then second to those with a diagnosis. This issue does not seem to be explained in the methods section. More importantly, it is well-understood that selection based on a common effect of the exposure (COVID-19) and outcome will distort the association between exposure and outcome seen in the selected sample ('collider bias'). It is important that this issue is fully addressed in any revised version of the paper.
4. Patients are followed from index dates corresponding to their positive or negative SARS-CoV-2 test. Does this mean that follow up does not account for the calendar date on which events occurred. If so, this issue should be addressed in revised analyses, given the dramatic fluctuations over calendar time in the incidence of COVID-19 and of PASCs (for example because of restrictions to health services during periods of lockdown).
5. A disadvantage of IPT weighting is that extreme weights can make results unstable. As the

authors note, the IPTW is a function of the propensity score. Please repeat analyses by controlling for a (nonlinear function of) the propensity score, and confirm that results are similar or address any discrepancies.

Minor comments

6. I do not understand why the authors make separate Bonferroni adjustments for diagnoses and medications. Surely they are all reported if "positive", so there should be a single Bonferroni adjustment for all? And why did they exclude negative hazard ratios? It would be interesting to know whether there were any events that were reduced after COVID-19.

7. The derivation of the propensity scores is described too briefly. Explain what is meant by "regularized logistic regression", the nature of the regularization parameters, and the cross-validation pipeline.

8. In the subgroup analyses, did the authors re-estimate the propensity scores and IPT weights? If not, is it valid to assume that they apply similarly across subgroups?

9. Dichotomizing numerical variables such as BMI (this was dichotomised at 40) can lead to residual confounding. Please increase the number of categories, or model nonlinear effects (eg using splines).

10. Typo on page 34: "3 or o digits"

Reviewer #3 (Remarks to the Author):

The article "Understanding Post-Acute Sequelae of SARS-CoV-2 Infection through Data-Driven Analysis with Longitudinal Electronic Health Records: Findings from the RECOVER Initiative", from Zang and colleagues performs a hypothesis free exploration of the post-acute sequela of SARS-CoV-2, an important and still poorly understood aspect of the infection, affecting large number of individuals. The combined exploration of both diagnosis and medications, grouped by organ system, is a nice approach. In the presented research is clearly of high quality, with a lot of effort put in to it, so well done. Some interesting findings are presented which could be further promoted with some adjustments.

To me it seems like your strongest results are those which replicate across the two cohorts, and I would advice that you highlight these better by restructuring your results section to focus on those findings, currently these are not presented as main findings in the abstract. It seems a shame not to highlight these if this is indeed the largest study examining this to date, a replication of results between two different sites seems like a strong finding to me.

Its unclear from the current presentation why the INSIGHT cohort is presented as the main sample, other than the fact that it has more samples overall who fulfill the study conditions and also test positive, but on the other hand the OneFlorida+ sample has a larger starting population.

In figure 2. It would be beneficial to highlight which conditions and medications replicated across the two cohorts. It would also be useful to see the incidence numbers for diagnosis/medication for context to the aHR (this would be useful in figure 4 as well).

In figure 3. It would be useful to have the sample sizes of the various groups being compared along the top x-axis and similarly the number of observed diagnosis along the y-axis.

Could you elaborate your thoughts on the high excess cumulative incidence rate of the healthy population for Dyspnea (70.1) and Diabetes (12.1)

You hint in the discussion that a reason for the differences observed between the cohorts, could be caused by differences in vaccination uptake. Would it be possible to adjust for vaccination status in

your analysis? And if not could you further present in the discussion if this data was not available, or the reason for not including it.

In general I think it would be useful to clarify where the start end dates of the study sits in terms of the wider pandemic context in the US i.e. infection waves and/or vaccination roll-out, in the two regions examined, as this would provide good context for the presented results. You do have a section that discuss this to some extent, but if you could overlay the vaccine roll out data in your extended data fig1 that would make this super clear.

Minor.

Clarify in panel (A) of figure 1, top box. Its not 100% clear what this box presents, I believe its number of patients until the end of 2021? Or until start November 2021?

Consider, rephrasing/changing your argumentation, about previous studies sample sizes. This sentence from your discussion makes it sounds like you rely on the systematic review alone to assess the sample size of the earlier studies you have just listed, which I don't think is the case. "Additionally, according to a recent systematic review¹, most of these studies are small (less than 1,000 patients)."

Great that you have shared your analysis code for replication, however it would be useful with even a minimal README file in your git repository to help an outsider navigate your repository.

Review Response

We highly appreciate the constructive comments and suggestions from reviewers. In the following, we respond to them point by point.

Reviewer #1:

1. This is an interesting real world electronic health record study of events recorded in the health care records of two very large datasets in New York and Florida between 31 and 180 days after COVID-19 compared to individuals who never tested positive over the study period. Advantages are the study size and the attempt to use inverse probability score weighting to account for clear evident differences between groups. None of the findings are particularly surprising in the context of the international literature published so far, with the exception of the fact that the rates overall of pasc are really quite low – even dyspnoea is reported in only 8% of hospitalised cases after 180 days. The fact that access issues may undermine the reporting in the health record is important, making interpretation difficult without a benchmark using population-based data. Indeed, it is disappointing to note that there is no reference to other large scale electronic health record datasets or population based samples, which are already published. I'm interested that in the conclusion the authors states these data had exhibited "high incident risks". I think their overall incident data presented are relatively low compared to other datasets, unless I have misunderstood the data presented.

Response: Thanks for your comments. We apologize for the confusion. The followings are clarifications for the questions raised.

First, the numbers shown in the heatmap of Fig 4 are **excess burdens** per 1000, which is calculated by the difference of the adjusted cumulative incidence in the COVID-19-positive patients versus the negative patients in the corresponding group. For example, the number 89.9 for dyspnea in hospitalized patients means cumulatively there are 89.9 per 1000 (or 8.99%, after adjustment for baseline covariates) **more** patients have incident dyspnea for COVID-19 positive patients who were hospitalized in the acute infection phase (within 30 days after COVID-19 confirmation) compared to COVID-19 negative patients who were hospitalized in the same time period, not the absolute incidence. We have updated the caption of Fig 4 to explicitly clarify that the numbers in the figure are excess burdens and how they are calculated. In addition, we further added Extended Fig. 4 and 5 to show the absolute cumulative incidence of PASC across different subgroups in INSIGHT and OneFlorida+, where we can see that cumulatively 20.61% and 21.04% of COVID-19 patients

who were hospitalized in the acute phase got incident dyspnea in the post-acute phase within the INSIGHT and the OneFlorida+ cohorts. We have also added a lot of discussions on relevant references in both the introduction and discussion sections. Specifically, for the results reported from the US Veterans Affairs (VA) Research^{1,2}, the excess burden of dyspnea is 7.68% for patients hospitalized in the acute phase² (see Fig 6 and Supplementary Table 4 in **Ref 2**, where shortness of breath is associated with number 76.83 per 1000, which is comparable to 89.9 per 1000 in our study considering the population difference).

Second, regarding data access, as we have provided in the data availability statement, the INSIGHT data can be requested through <https://insightcrn.org/>, and the OneFlorida+ data can be requested through <https://onefloridaconsortium.org>. Both data are HIPAA-limited and thus data use agreements can be established with the INSIGHT and OneFlorida+ networks to gain access.

Third, the added value of this paper is that 1) we validated the conclusions from prior EHR-based studies that PASC is a collection of symptoms and conditions involving multiple organ systems; 2) we demonstrated the commonality and heterogeneity of PASC across different geographical areas with distinct characteristics of patient socioeconomic status, hitting waves in terms of SARS-CoV-2 variants, availability of vaccines, etc. For example, we found many consistent PASC-related diagnoses in both NYC and Florida, including dementia, hair loss, pressure ulcers, pulmonary fibrosis, dyspnea, pulmonary embolism, chest pain, abnormal heartbeat, malaise, and fatigue, and there are more potential PASC conditions identified from the patients in NYC than Florida, such as myopathies, thromboembolism, fluid disorders, etc. (details see Fig. 2). The adjusted hazard ratios of these potential PASC conditions were also higher in INSIGHT compared to OneFlorida+ (Fig. 3).

2. Overall, while not exactly novel in terms of findings, this is an interesting contribution which validates existing publications – in particular, increased risk of thromboembolism and lung fibrosis, but also many non-specific conditions. I was interested that they did not report increased incidence of for example new myocardial infarction or new diabetes diagnosis, which has been found in other datasets. Indeed, it was disappointing to see only positive results reported. Were there any interesting exceptions here from existing literature?

In the population statistics section, would be useful to know how representative of the general population people registered with these patient centred clinical research networks are. Do they just include people with access to health insurance

Response: Thanks for the comments. We would like to provide the following clarifications to these questions.

First, similar to existing studies^{1,2}, the goal of our study is to identify potential PASC conditions from large EHR cohorts. Although we did not find any surprising exceptions, our findings validated many conclusions from prior studies, which is important as different cohorts have different characteristics (e.g., the VA studies^{1,2} were based on the veteran population while our study was based on the civilian population). Moreover, one unique aspect of our study is we have incorporated two large and geographically distinct EHR cohorts, in this way we can compare the commonality and heterogeneity of PASC across geographically different populations, as we have responded to the previous question.

Second, as shown in the updated Sensitivity Analysis Section, myocardial infarction was identified in INSIGHT (aHR 1.29, 95% CI [1.13, 1.48], Pvalue=0.000208) but was not qualified in OneFlorida+ (aHR 0.96, 95% CI [0.80, 1.15], Pvalue=0.6407). As shown in updated Figure 3, diabetes was identified in INSIGHT (aHR 1.27, 95% CI [1.17, 1.37], Pvalue= 8.4×10^{-9}) but not significant in the OneFlorida+ (aHR 1.11 95% CI [1.00, 1.23], Pvalue=0.04). Both conditions are great examples showing the heterogeneity of PASC. There could be multiple hypotheses on the reasons behind such heterogeneity, such as the older age of INSIGHT patients, the higher acute phase severity of the ancestral wave, and the lower socioeconomic status of the OneFlorida+ patients, which have been discussed in more detail in the Discussion Section.

Third, our EHR data covered all patients who had records in any sites in INSIGHT or OneFlorida+ (See Data Section), not requiring if they had insurance coverage or not. We further compared our two cohorts with general PCORnet patients and CDC jurisdictions, according to patient demographics. The PCORnet data and CDC data were adapted from the supplementary table in https://stacks.cdc.gov/view/cdc/113252/cdc_113252_DS1.pdf. As shown in the following table, both INSIGHT and OneFlorida+ cohorts were fairly representative of the national population but covered more older, black, and hispanic patients, which reflected the characteristics of the residents in New York City and Florida areas.

[REDACTED]

3. What are the statistics in the results section - OR HR – it took me till the methods to get what you'd done, due to the English in the description. Eg "We reported their incident risks in the adjusted hazard ratio (Fig. 2) and excess burdens in the adjusted excess cumulative incidence" should probably be "Incident risks are reported as adjusted hazard ratios (Fig. 2) and adjusted excess cumulative incidences over 180 days."

On this point about excess cumulative incidences : further amendments to English are needed in the paragraphs discussing these subgroup analyses. I am also not entirely sure that you are presenting the "excess cumulative incidence" in Figure 3, or the cumulative incidences by group. You state "We estimated the adjusted cumulative incidence of each potential PASC diagnosis per 1,000 patients at 180 days in different groups and compared the excess burden of it¹⁸, which is the difference between the adjusted cumulative

incidences of a specific diagnosis in the SARS-CoV-2 infected subpopulation and the corresponding control subgroup” What therefore is provided in subjects overall? It looks like from many of the figures reported are cumulative incidence rather than the difference, or excess, as the comparative subgroups often straddle the overall, (but this is not uniformly the case, eg cognition and dementia). It’s also really important that you label this **as per 1000 Patients**, in the legend, and preferably by the scale, which seems to imply a 100 top value, which is misleading in my view.

Response: Thanks for your comments. Throughout the paper, we have leveraged three methods to quantify the incident risk of potential PASC conditions, including hazard ratio (modeled by the Cox proportional hazard model), cumulative incidence (modeled by the Aalen-Johansen model), and excess burden (the difference between cumulative incidences in different exposure groups. In the following, we provide more details on these quantities and how we modified the text to make them clear.

First, we reorganized the content in results so the results from INSIGHT (Fig 2) were presented first, and the results from OneFlorida+ (Fig 3), followed by stratified analysis (Fig 4). Now in Fig 2 and 3, we have provided both the *adjusted hazard ratio* and *adjusted cumulative incidence per 1000 patients* to quantify the risks of each condition. We have revised the text and the figure captions to explicitly describe these definitions. We didn’t use odds ratio (OR) in our study as all our analyses were conducted in time-to-event settings.

Second, for the results of the stratified analysis shown in Figure 4, we adopted excess burden per 1000 patients (difference of adjusted cumulative incidence in SARS-CoV-2 infected patients versus non-infected patients across exposure groups), with the goal of highlighting the differences across different subgroups. In the revised text, we clearly presented in the 1st paragraph of the Stratified Analysis Section “Here we quantified PASC risk by the excess burden per 1,000 patients, which is defined as the difference between the adjusted cumulative incidences of a specific condition in the SARS-CoV-2 infected patient population versus the corresponding control population in a particular group”.

4. The causal inference procedure accounted for the population differences between those who tested positive and those who did not, using inverse propensity score weighing, (at least for the Hazard ratios, but I am not certain about the cumulative incidences. Therefore findings should not be influenced by differences in populations testing positive. However, these differences are quite marked, and did make me think throughout that I would prefer not to have seen the words “causal inference” used so shamelessly. Ultimately this is a large electronic health data study with some clever stats, but its very clearly observational and I would not make causal inferences from it.

This is further underlined by the fact that the two cohorts had quite differing baseline characteristics between the positive and negative groups, which it appears may then play out in the results.

Response: Thanks for the comments and suggestions. We have removed all claims about “causal inference” and changed them to observational study with propensity score adjustment. It is also true that our conclusion about the incidence risk of potential PASC changed over different populations, as evidenced by the difference in results obtained from INSIGHT and OneFlorida+. In addition, we have added adjusted cumulative incidence in both the covid positive and negative groups in Fig. 2 and 3, where the cumulative incidence was estimated using the Aalen-Johansen model and adjusted by the same stabilized IPT weights as used in calculating the adjusted hazard ratio.

5. Reading through I found it difficult to work out the differences in timings of infection between the two cohorts, and the supplementary figure was helpful and confirmatory. Table one points out the time periods when data was collected and there are interesting differences in the time periods for the majority of positive cases between the two cohorts. This then may be reflected in the descriptive data and the differences in subsequent findings in the cohorts. This as well as the other cohort differences alluded to above make interpretation difficult and the study purely descriptive. The authors do attempt to discuss this in relation to differences between variants and in the advent of vaccination. However, with such a large dataset it is a shame not to segment into variant / vaccination groupings.

Response: Thanks for the comments and suggestions. We have made the following changes in the revised version.

First, we have updated Extended Data Fig 1 to illustrate the temporal trends of infections in both cohorts and more contextual information including periods of variants and vaccinations.

Second, in the Stratified Analysis Section, we have added a paragraph summarizing the risk of PASC according to excess burden within different waves of SARS-CoV-2, with details provided in Extended Data Fig. 6.

Third, The vaccination started in early December 2020 (see Extended Data Fig. 1), and more than half of the SARS-CoV-2 infections happened before the vaccine was available. For patients after December 2020, only 2% of the population in the covid positive group and 5% in the covid negative group had baseline vaccination information due to the challenge of capturing them in the EHR, which has been acknowledged as one of the limitations of this study in the Discussion Section.

6. For the international reader some context on the ADI would be helpful. What do the differences between the two cohorts mean (need to know what the range and interpretation is).

Response: Thanks for the comments. The Area Deprivation Index (ADI) ranks neighborhoods by their socioeconomic disadvantage in the nation, ranging from 1 to 100, with 1 and 100 indicating the lowest and highest level of disadvantage respectively. We have added these descriptions of ADI in both the Result - Population Statistics Section and Method – Group assignment and baseline covariates section.

Reviewer #2:

The authors examine the association of COVID-19 with 137 diagnoses and 459 medications (I think there is a typo when they say 359 elsewhere in the paper) in the post-acute period, which they define as from 30 to 180 days after COVID-19 diagnosis. They refer to these collectively as post-acute COVID-19 sequelae (PASCs). Their analyses are based on a “high-throughput causal inference pipeline”, which is implemented in two electronic health record (EHR) datasets. Only diagnoses and medications with (1) adjusted hazard ratios larger than 1, (2) P-value smaller than 3.6×10^{-4} for diagnoses and 1.4×10^{-4} (for medications) (the authors made a Bonferroni method for multiple testing) were retained as potential PASCs. The reported results are mainly based on the larger INSIGHT cohort (~35k SARS-CoV-2 positive, ~326k negative). The authors report that results are rather different in the smaller OneFlorida+ cohort (~22k positive, ~177k negative).

Major comments.

1. The wide range of post-COVID events analysed is a strength of the paper. However, it means that in the results section there is a strong emphasis on the minority of associations that were identified as PASCs. (1.1) We have no information on the specificity of the identified associations, because the non-selected results are, as not reported in detail. The authors should include information about the diagnoses and medications that were not identified as PASCs. (1.2) They should also comment on the role of the number of events, because if there is a small number of events the association has to be very large to achieve a p value smaller than the threshold. It is likely that many potential PASCs with large HRs were not considered positive. (1.3) The Bonferroni correction is based on an assumption that the joint null hypothesis is true – this is not plausible for post-COVID events.

Response: Thanks for the comments and suggestions. We have made the following updates in this revised version.

First, we have provided detailed statistics of all diagnoses and medications, including both selected ones and unselected ones, in Supplementary Tables 3 and 4. In addition, we have also reported how the selected PASC conditions will change when using different screening criteria in the Sensitivity Analysis Section and Extended Data Fig. 7.

Second, we have added the adjusted cumulative incidences in both the COVID positive and negative patient groups in Figs 2 and 3. In our investigation, we required at least 100 identified incidences in the COVID-positive group, to add more power to the final estimated effects and to overcome the large HR ratio but small sample size concern as mentioned. We also discussed lifting the threshold 100 in the sensitivity analyses section. Thanks to the large sample sizes of both cohorts, all of our identified likely PASC conditions showed a large sample size.

Finally, to reduce the chance of false positive discovery, we adopted the Bonferroni method (BF) controls the familywise error rate (FWER), namely the probability of making one or more false discoveries, which is a very stringent method. However, we still found a significant number of signals as seen in Fig. 2-3. Furthermore, we added sensitivity analysis by considering the false discovery rate-based method (FDR) – Benjamini-Yekutieli method (BY)³ – which requires no assumptions about the correlations of different tests (possibly more suitable than Benjamini-Hochberg procedure and Storey’s q-value method in the PASC setting). By using the BY method, in addition to our identified PASC conditions reported in the main text, we found other 7 and 1 potentially significant PASC diagnoses in the INSIGHT the OneFlorida+, respectively. See our results in the sensitivity analyses section.

2. The same issue applies to subgroup analyses. Even if associations were identical within subgroups, with the same hazard ratio as overall, they could be designated as not associated because the significance within subgroups no longer reaches the threshold.

Response: Thanks for the comment. In our Fig. 4 and extended figure 2, we further highlighted significant subgroup conditions that had aHR’s P-value $< 8.39 * 10^{-5} = 0.05 / (137 + 459)$.

3. The authors have implemented a “causal inference” pipeline in which they aim to emulate a target trial. However the flow chart in Figure 1 implies that there are two stages of selection before patients enter the target trial: first the restriction to those who tested, then second to those with a diagnosis. This issue does not seem to be explained in the methods section. More importantly, it is well-understood that selection based on a common effect of the exposure (COVID-19) and outcome will distort the association between exposure and outcome seen in the selected sample (‘collider bias’). It is important that this issue is fully addressed in any revised version of the paper.

Response: Thanks for this important comment. In the following, we clarify our thoughts and the modifications we made in this revised version.

We have deleted all claims about “causal inference” to avoid confusion as our study is a retrospective analysis of observational EHR cohorts. We required SARS-CoV-2 PCR test records to construct infected and non-infected patient groups, as well as at least one documented diagnosis record in the baseline to capture their underlying conditions. We further required at least one documented diagnosis record in the post-acute phase (+31 to +180 days), because: (a) patients were required to be alive after their acute phase, and (b) to capture enough information in the post-acute phase; which were critical to study the post-acute sequelae of SARS-CoV-2 and is required by the nature of the definition of post-acute sequelae (selection in the follow-up period). These are common settings adopted by existing Long Covid literature based on observational EHR data.^{1,2,4} In addition, the collider bias is a problem for the generalisability from observational data to a more general population⁵, which remains an open challenge in the COVID-19-related studies⁵. That is a major advantage of our study of using two large-scale and distinct EHR cohorts, which can identify more generalizable PASC knowledge.

4. Patients are followed from index dates corresponding to their positive or negative SARS-CoV-2 test. Does this mean that follow-up does not account for the calendar date on which events occurred. If so, this issue should be addressed in revised analyses, given the dramatic fluctuations over calendar time in the incidence of COVID-19 and of PASCs (for example because of restrictions to health services during periods of lockdown).

Response: Thanks for the comment. The period of the index calendar date (March 2020 – June 2020, July 2020 – October 2020, November 2020 - February 2021, March 2021 – June 2021, July 2021 – November 2021) was included as baseline covariates to account for the potential impact on the different stages of the pandemic (See Table 1 and Method -- Group assignment and baseline covariates section). We also added discussion on the different waves (corresponding to different variants of SARS-CoV-2) in stratified analysis and Extended Data Fig 6. The fluctuations over calendar time were illustrated in Extended Data Fig 1.

5. A disadvantage of IPT weighting is that extreme weights can make results unstable. As the authors note, the IPTW is a function of the propensity score. Please repeat analyses by controlling for a (nonlinear function of) the propensity score, and confirm that results are similar or address any discrepancies.

Response: Thanks for the comment. We have updated our re-weighting method and repeated our analyses by further controlling for extreme weights. In our previous manuscript, we adopted stabilized weights $w = \frac{Z * P(Z=1)}{P_{\theta}(Z=1|X)} + \frac{(1-Z) * P(Z=0)}{1 - P_{\theta}(Z=1|X)}$ ^{6,7}, which can control

for the large weight and inflated re-weighted population size. In the revised manuscript, we further trimmed stabilized IPTW which was either smaller than 1st or larger than 99th percentiles in each emulation.⁶ Same as our previous results, all the baseline covariates were balanced by using these trimmed and stabilized weights. Taking the sample weights distribution of 137 emulations for diagnoses as an example, see below:

Diagnosis Distributions^a	Propensity Score	Stabilized IPTW	Stabilized and trimmed IPTW
Count^b	27449382	27449382	27449382
mean	0.167	0.998	0.971
std	0.130	0.558	0.277
min	0.003	0.180	0.318
25%	0.048	0.855	0.855
50%	0.150	0.939	0.939
75%	0.240	1.055	1.055
max^c	0.927	29.898	2.617

Medication Distribution^a	Propensity Score	Stabilized IPTW	Stabilized and trimmed IPTW
Count^b	94494774	94494774	94494774
mean	0.167	0.998	0.971
std	0.130	0.558	0.277
min	0.004	0.180	0.320
25%	0.048	0.855	0.855
50%	0.150	0.939	0.939
75%	0.240	1.055	1.055
max^c	0.928	29.195	2.515

- We reported distributions of the estimated propensity score, stabilized inverse treatment weight, and stabilized and trimmed inverse treatment weights from all the emulations. The distributions were quantified by the mean, standardized difference, and quantiles.
- The total number of patients involved in the high-throughput emulations; 137 emulations for diagnosis-specific cohorts, and 459 emulations for the medication-specific cohorts. Patients can be used for multiple emulations.
- We highlighted the maximum weights in red.

The largest weight in our previous stabilized IPTW approach was 29.898; after controlling for extreme values by trimming, the largest weight now is 2.617. We observed the same phenomenon in the medication table. A smaller IPTW led to less variability (smaller confidence interval and smaller P-value). We thus identified more potential PASC conditions for further analysis. We have updated our main results in Fig 2-3 and associated descriptions and discussions accordingly. Our major findings still hold – more PASC diagnoses and a higher risk of PASC in NYC than in Florida, and only dementia, hair loss, pressure ulcers, pulmonary fibrosis, dyspnea, pulmonary embolism, chest pain, abnormal heartbeat, malaise, and fatigue, were consistently identified across two population – highlighting the heterogeneous risks of PASC in different populations.

Minor comments

6. I do not understand why the authors make separate Bonferroni adjustments for diagnoses and medications. Surely they are all reported if “positive”, so there should be a single Bonferroni adjustment for all? And why did they exclude negative hazard ratios? It would be interesting to know whether there were any events that were reduced after COVID-19

Response: Thanks for the comment. We have repeated our analyses by using a consistent p-value threshold of $0.05/(137+459) = 8.39 \text{ e-}5$, where a class of tests is the sum of 137 diagnosis tests and 459 medications. The complete results on all diagnoses and medications for INSIGHT have been provided in Supplementary Table 3, where we did not find any significant negative conditions (aHR < 1). We have added these results in the sensitivity analysis section.

7. The derivation of the propensity scores is described too briefly. Explain what is meant by “regularized logistic regression”, the nature of the regularization parameters, and the cross-validation pipeline.

Response: Thanks for the comment. We have added the following sentence in the Method – Inverse propensity score weighting for adjustment section. “We used logistic regression with the L2 penalty term for PS calculation, with the optimal regularization strength determined through grid search over hyper-parameter space (10^{-2} , $10^{-1.5}$, 10^{-1} , $10^{-0.5}$, 1, $10^{0.5}$, 10^1 , $10^{1.5}$, 10^2 , and no penalty).” The algorithm for selecting the best PS model is detailed in the Extended data Table 3.

8. In the subgroup analyses, did the authors re-estimate the propensity scores and IPT weights? If not, is it valid to assume that they apply similarly across subgroups?

Response: Thanks for the question. The answer is yes and we have added the following clarification sentence in the 1st paragraph in the Stratified analysis section., “for each subgroup analysis, we built the infected subgroup and its control subgroup and re-estimated the stabilized IPT weights for adjustment”. We further highlighted significant conditions (p-value < $8.39 \text{ e-}5$) across different subgroups in Fig. 4 and Extended Data Fig. 2.

9. Dichotomizing numerical variables such as BMI (this was dichotomised at 40) can lead to residual confounding. Please increase the number of categories, or model nonlinear effects (eg using splines).

Response: Thanks for the comment. We apologize for the confusion and would like to make the following clarification. Rather than dichotomizing BMI, we used the following five BMI categories as baseline covariates – “The body mass index (BMI) was categorized into underweight (<18.5 kg/m²), normal weight (18.5 kg/m² – 24.9 kg/m²), overweight (25.0 kg/m²– 29.9 kg/m²), obesity (>= 30.0 kg/m²), and missing according to the CDC guideline for adults²⁹ “(https://www.cdc.gov/healthyweight/assessing/bmi/adult_bmi/index.html). We have added the above description in the Method – Group assignment and baseline covariates section.

10. Typo on page 34: “3 or o digits”

Response: Thanks for the comment. We updated this to “3 or 4 visits” in the Method – Group assignment and baseline covariates section in our revised manuscript.

Reviewer #3:

The article “Understanding Post-Acute Sequelae of SARS-CoV-2 Infection through Data-Driven Analysis with Longitudinal Electronic Health Records: Findings from the RECOVER Initiative”, from Zang and colleagues *performs a hypothesis free exploration of the post-acute sequela of SARS-CoV-2, an important and still poorly understood aspect of the infection, affecting large number of individuals.* The combined exploration of both diagnosis and medications, grouped by organ system, is a nice approach. In the presented research is clearly of high quality, with a lot of effort put in to it, so well done. Some interesting findings are presented which could be further promoted with some adjustments.

1. To me it seems like your strongest results are those which replicate across the two cohorts, and I would advice that you highlight these better by restructuring your results section (highlight) to focus on those findings, currently these are not presented as main findings in the abstract. It seems a shame not to highlight these if this is indeed the largest study examining this to date, a replication of results between two different sites seems like a strong finding to me.

Response: Thanks for the suggestion, our research indeed showed the heterogeneity and commonality of PASC across different populations, Specifically, we found more PASC diagnoses and a higher risk of PASC in NYC than in Florida, and only dementia, hair loss, pressure ulcers, pulmonary fibrosis, dyspnea, pulmonary embolism, chest pain, abnormal heartbeat, malaise, and fatigue, were replicated across two population. Our analyses highlighted the heterogeneous risks of PASC in different populations. In the revised manuscript, we have 1) reorganized the overall content structure to prioritize the Comparison with one OneFlorida+ Cohort section ahead of the Stratified analysis section,

to highlight the differences in PASC risk and discuss more on those replicated across two cohorts; 2) highlighted the abovementioned heterogeneous risks and replicated findings in all the main Figures (Figs 2, 3, and 4), Abstract, and Introduction Sections.

2. Its unclear from the current presentation why the INSIGHT cohort is presented as the main sample, other than the fact that it has more samples overall who fulfill the study conditions and also test positive, but on the other hand the OneFlorida+ sample has a larger starting population.

Response: Thanks for the comments. In the revised version, we have reorganized the presentation and tried to make two cohorts as parallel studies and compared their commonality and heterogeneity in terms of the PASC risks, which were highlighted in Fig 3 and discussed in the Results from the OneFlorida+ cohort Section.

3. In figure 2. It would be beneficial to highlight which conditions and medications replicated across the two cohorts. It would also be useful to see the incidence numbers for diagnosis/medication for context to the aHR

Response: Thanks for the suggestion. We have marked replicated diagnoses/medications across two cohorts in Fig. 2 by ‡ symbols. We further compared the two datasets in detail in Fig. 3 and the Results from the OneFlorida+ cohort section.

4. In figure 3. It would be useful to have the sample sizes of the various groups being compared along the top x-axis and similarly the number of observed diagnosis along the y-axis.

Response: Thanks for the suggestion. We have revised Fig. 4. (previously Fig. 3) by adding the percentage of the subgroup population at the top of the figure, highlighted replicated conditions by ‡ symbols, and also highlighted subgroup risk with a significant p-value after correction ($< 8.39 * 10^{-5}$) with red squares.

We have also added Extended Data Fig. 4 to show the absolute cumulative incidence in each subgroup. We also revised figures for OneFlorida+ in Extended Data Fig 2 (Fig. 4's counterpart) and Extended Data Fig. 5 (Extended Data Fig. 4's counterpart).

5. Could you elaborate your thoughts on the high excess cumulative incidence rate of the healthy population for Dyspnea (70.1) and Diabetes (12.1)

Response: Thanks for the comment. "Even for healthy patients without documented baseline Elixhauser comorbidities, we observed incident dyspnea, pulmonary fibrosis, and

chest pain burdens in both two cohorts, and diabetes burden in the INSIGHT cohort.” We added these results to the Stratified Analysis Section. The potential implication is, even for healthy patients, if they get SARS-CoV-2 infected, they might develop these conditions in the post-acute infection period.

6. You hint in the discussion that a reason for the differences observed between the cohorts, could be caused by differences in vaccination uptake. Would it be possible to adjust for vaccination status in your analysis? And if not could you further present in the discussion if this data was not available, or the reason for not including it.

Response: Thanks for the comment. First, we adjusted for different index periods by every quarter, aiming to control for potential temporal factors (vaccine, variants of concerns, etc.). Second, the vaccine began in early December 2020 (see the revised extended data fig 1), and more than half of the patients got infected before the vaccine was available. Third, even for patients after December 2020, the number of recorded patients was limited, please see the table below. We defined the fully vaccinated as two shots of mRNA vaccine (Pfizer, or Moderna) or one shot of J&J, see <https://www.cdc.gov/coronavirus/2019-ncov/vaccines/stay-up-to-date.html>. In all, the vaccine baseline covariates before the index date only accounted for 2% of the population in the covid+ group and 5% in the covid- group, barely changing the final screened conditions considering their significant hazard ratio. Studying how the vaccine influences the risk of long Covid is very interesting and left as a future study with a more sophisticated experiment design, and also relies on the ongoing efforts of cumulating and collecting more vaccination data (e.g. registry database, which is not available yet). We acknowledged this as a limitation in the discussion section.

	INSIGHT COVID+		INSIGHT COVID-	
	N	%	N	%
Total number	61305	100%	577174	100%
Fully vaccinated - Pre-index	811	1%	17229	3%
Partially vaccinated - Pre-index	823	1%	11753	2%
No evidence - Pre-index	59671	97%	548197	95%

7. In general I think it would be useful to clarify where the start end dates of the study sits in terms of the wider pandemic context in the US i.e. infection waves and/or vaccination roll-out, in the two regions examined, as this would provide good context for the presented results. You do have a section that discuss this to some extent, but if you could overlay the vaccine roll out data in your extended data fig1 that would make this super clear.

Response: Thanks for the suggestion. We have updated the Extended Fig. 1 with wider pandemic context information, including different waves of SARS-CoV-2 variants, and the vaccination dates. We highlighted three waves covered by our datasets: the 1st wave was

dominated by the ancestral strain, the 2nd wave was a mixture of Alpha and others, and the 3rd wave was dominated by the Delta variant. The earliest public covid vaccine began in early December 2020. We further added stratified analysis on Different Waves in the Stratified Analysis Section.

Minor.

8. Clarify in panel (A) of figure 1, top box. Its not 100% clear what this box presents, I believe its number of patients until the end of 2021? Or until start November 2021?

Response: We have revised Fig. 1 for better clarification. The top box in the revised Fig. 1 shows the number of patients who took any COVID PCR/Antigen tests from March 1st, 2020 to November 30, 2021.

9. Consider, rephrasing/changing your argumentation, about previous studies sample sizes. This sentence from your discussion makes it sounds like you rely on the systematic review alone to assess the sample size of the earlier studies you have just listed, which I don't think is the case. "Additionally, according to a recent systematic review¹, most of these studies are small (less than 1,000 patients)."

Response: Thanks for the comment. We have deleted those small sample size-related claims in this revision and we have changed the particular sentence mentioned in this comment to "All these studies investigated a single dataset." in the first paragraph of the Discussion section,

10. Great that you have shared your analysis code for replication, however it would be useful with even a minimal README file in your git repository to help an outsider navigate your repository.

Response: thanks for your great suggestion. We have added a README file in our git repository https://github.com/calvin-zcx/pasc_phenotype, including an introduction to the NIH RECOVER project, related work, system requirements, how to set up the python environment, code structure, and associated shells to run codes.

References

1. Al-Aly, Z., Xie, Y. & Bowe, B. High-dimensional characterization of post-acute sequelae of COVID-19. *Nature* **594**, 259–264 (2021).
2. Xie, Y., Bowe, B. & Al-Aly, Z. Burdens of post-acute sequelae of COVID-19 by severity of acute infection, demographics and health status. *Nat Commun* **12**, 6571 (2021).
3. Benjamini, Y. & Yekutieli, D. The Control of the False Discovery Rate in Multiple Testing under Dependency. *The Annals of Statistics* **29**, 1165–1188 (2001).
4. Cohen, K. *et al.* Risk of persistent and new clinical sequelae among adults aged 65 years and older during the post-acute phase of SARS-CoV-2 infection: retrospective cohort study. *BMJ* **376**, e068414 (2022).
5. Griffith, G. J. *et al.* Collider bias undermines our understanding of COVID-19 disease risk and severity. *Nat Commun* **11**, 5749 (2020).
6. Austin, P. C. & Stuart, E. A. Moving towards best practice when using inverse probability of treatment weighting (IPTW) using the propensity score to estimate causal treatment effects in observational studies. *Statistics in Medicine* **34**, 3661–3679 (2015).
7. Cole, S. R. & Hernán, M. A. Constructing Inverse Probability Weights for Marginal Structural Models. *American Journal of Epidemiology* **168**, 656–664 (2008).

REVIEWER COMMENTS

Reviewer #1 (Remarks to the Author):

I congratulate the authors for carefully and thoughtfully taking account of all the reviewers comments, and addressing them in the new manuscript. The resulting work is, in my view, a comprehensive and novel addition to understanding of PASC and worthy of publication in nature communications

Reviewer #2 (Remarks to the Author):

The authors have made a number of changes in light of the reviews of their original manuscript. I have the following comments on the revised manuscript.

Major comments

1. Comparison of results between the two cohorts.

The authors need to be more careful in their comparison of the two cohorts, because selecting associations based on p-value thresholds has a number of consequences. First, selection implies that the association in an identical replication cohort will be smaller on average than the selected association (statistically, this is known as shrinkage). The more that chance findings are being selected, the larger the differences between the associations in the discovery and replication cohorts. Therefore, the pattern of much smaller associations in OneFlorida than INSIGHT seen in Figure 3 may imply that a number of the INSIGHT associations are chance findings. Second, the smaller sample size in OneFlorida than INSIGHT implies that associations in OneFlorida have to be larger in order to meet the p-value threshold. Therefore, it is unsurprising that fewer associations in OneFlorida than INSIGHT met the p value threshold. Third, the size of hazard ratio required to meet the p value threshold decreases as the cumulative incidence of the event increases. This is what we see in Figure 2, in which the CIF tends to increase as the magnitude of the HR decreases.

In light of these considerations, I suggest that the authors reword or provide appropriate interpretation of the following statements, and similar statements elsewhere in the manuscript. They should ensure that the issues discussed above are addressed in the discussion section of their manuscript.

Lines 39-40: "We found more PASC diagnoses and a higher risk of PASC in NYC than in Florida..."

Lines 42-43: "Our analyses highlight the heterogeneous risks of PASC in different populations."

Lines 83-84: "These results highlighted the potential heterogeneity of PASC over different patient populations and the need for replication studies before robust conclusions about PASC can be made."

Paragraph in the discussion: "We observed clear heterogeneity after replicating..."

The presentation of the findings in Figures 2 and 3 (and the results and discussion sections) seem to indicate confusion between two possible approaches to interpreting the results across the two cohorts. It is common (for example in analyses of genome-wide association studies) to consider the first dataset as "discovery" and the second as "replication", with associations meeting significance criteria in the discovery dataset only considered confirmed when replicated in the second dataset. The following statement (lines 40-42) implies that this is what the authors intend: "... conditions including dementia, hair loss, pressure ulcers, pulmonary fibrosis, dyspnea, pulmonary embolism, chest pain, abnormal heartbeat, malaise, and fatigue, were replicated across two populations." However, they do not seem to interpret Figure 3 as demonstrating lack of replication for most conditions, and their description of the INSIGHT results implies that they consider all these to be confirmed in light of meeting the p value threshold within INSIGHT. It is

unclear what they believe about the OneFlorida associations, which are discussed much more briefly. Similarly, the second paragraph of the discussion seems to focus on the INSIGHT results, although it does not mention which dataset results are being described.

Alternatively, the authors may be interested in (line 30) “the heterogeneous risks of PASC in different populations”. In this case, they should present the results that met the p value threshold from each dataset, instead of focussing on the INSIGHT results (Figure 2, lines 150 to 210) and then comparing the results for post-acute sequelae identified in INSIGHT with those from OneFlorida. In any case, Figure 3 reproduces the INSIGHT results presented in Figure 2: it makes little sense to include the same INSIGHT results for post-acute sequelae in each figure. It is unclear why the authors compared results for post-acute sequelae but not medications between the cohorts.

2. Subgroup analyses.

In their revised description of the subgroup analyses the authors focus on comparing burdens between subgroups. However, relative effects that are constant between a higher and lower burden setting will lead to a larger number of excess cases in the higher than lower burden setting. The authors should comment on how relative effects, as well as excess burdens, vary between subgroups.

3. Different waves.

The evolution from ancestral to alpha to delta to omicron waves was accompanied by the availability of primary course vaccination and then booster vaccination, which provided successively greater protection against severe COVID-19. This should be accounted for when comparing the excess burden of PASCs over different waves.

4. Causal inference

The authors say that they have “deleted all claims about “causal inference” to avoid confusion as our study is a retrospective analysis of observational EHR cohorts.” However, in various places they continue to refer to having used a target trial approach. Since the use of target trials explicitly aims to make causal inferences, references to target trials should also be deleted.

5. Inverse probability weighting.

Throughout, the authors refer to “inverse propensity score weighting”. Although the weights are based on the propensity score, they are not the inverse of the propensity score. They are inverse probability weights where the probability is that of being a case, as is made clear in the formula provided in the methods section. The wording should be changed throughout the manuscript.

The authors have trimmed the stabilised weights in order to avoid instability in estimates. However, this corresponds to not fully controlling the confounding. I would have preferred them to conduct a sensitivity analysis conditioning on a nonlinear function of the propensity score, which avoids this problem, but they did not take up this suggestion. There isn't much advantage to IPW when we are only dealing with baseline confounding.

Minor comments

Line 57: “... or result in biased findings”. This is vague – either clarify or delete.

Lines 57-58: “... prior studies have typically been conducted on specific populations”. There are quite a few general population studies and these should be cited.

Typo in the paragraph on sensitivity analysis: “potentially significant PASC diagnoses in the INSIGHT the OneFlorida+”.

Methods: “examined a total of 596 incident diagnoses (Supplemental Table 2) and medication

use...". Should this be "

Methods: "In addition, periods (March 2020 – June 2020, July 2020 – October 2020, November 2020 - February 2021, March 2021 – June 2021, July 2021 – November 2021) of the index date were used to account for potentially different stages of the pandemic." Given the rapid changes in incidence of COVID with calendar time this may not sufficiently account for confounding by calendar time. Instead, use a continuous function of calendar time such as a cubic spline.

Reviewer #3 (Remarks to the Author):

I am happy with the changes made by the authors in relation to my previous comments, and support the publication of this work.

We appreciate the thoughtful comments from the reviewers and provide our point-by-point response in this revision as below.

REVIEWER COMMENTS

Reviewer #1 (Remarks to the Author):

I congratulate the authors for carefully and thoughtfully taking account of all the reviewers comments, and addressing them in the new manuscript. The resulting work is, in my view, a comprehensive and novel addition to understanding of PASC and worthy of publication in nature communications

Reviewer #2 (Remarks to the Author):

The authors have made a number of changes in light of the reviews of their original manuscript. I have the following comments on the revised manuscript.

Major comments

1. Comparison of results between the two cohorts.

The authors need to be more careful in their comparison of the two cohorts, because selecting associations based on p-value thresholds has a number of consequences.

First, selection implies that the association in an identical replication cohort will be smaller on average than the selected association (statistically, this is known as shrinkage). The more that chance findings are being selected, the larger the differences between the associations in the discovery and replication cohorts. Therefore, the pattern of much smaller associations in OneFlorida than INSIGHT seen in Figure 3 may imply that a number of the INSIGHT associations are chance findings.

Second, the smaller sample size in OneFlorida than INSIGHT implies that associations in OneFlorida have to be larger in order to meet the p-value threshold. Therefore, it is unsurprising that fewer associations in OneFlorida than INSIGHT met the p value threshold. Third, the size of hazard ratio required to meet the p value threshold decreases as the cumulative incidence of the event increases. This is what we see in Figure 2, in which the CIF tends to increase as the magnitude of the HR decreases.

Response: Thanks for your comments. We would like to do the following clarifications.

First, our studies were based on two large EHR cohorts and we tried to control for chance findings by using stringent screening criteria (Method section) including 1) there were at least 100 events for any specific conditions, 2) the P-value of the associated aHR $< 8.39e-5$. We have further conducted extensive sensitivity analyses including using less stringent significance levels or lifting ≥ 100 event constraints, different covariates modeling methods, different PS calculation methods, etc. In this way, we hope to maximally reduce the probability that the differences or replicated PASC on two large EHR cohorts were chance findings.

Second, to further rule out the potential concern that different sample sizes could lead to different PASC risk patterns, we conducted a sensitivity analysis by (stratified) downsampling of the INSIGHT cohort to have exactly the sample number of patients as in the OneFlorida+ cohort for both the positive and negative groups (Sensitivity analysis section, and Extended Data Fig.11). As shown in the Extended Data Fig.11, after controlling for the population size, we still found consistent results as in our primary analysis (similar aHRs, and the replicated PASC conditions).

Third, chance findings can still exist due to the nature of observational data analysis. We have acknowledged this limitation in the discussions section, where we further emphasized our work as a hypothesis generation study and called for future biological mechanistic studies for the understanding of PASC.

In light of these considerations, I suggest that the authors reword or provide appropriate interpretation of the following statements, and similar statements elsewhere in the manuscript. They should ensure that the issues discussed above are addressed in the discussion section of their manuscript.

Lines 39-40: "We found more PASC diagnoses and a higher risk of PASC in NYC than in Florida..."

Lines 42-43: "Our analyses highlight the heterogeneous risks of PASC in different populations."

Lines 83-84: "These results highlighted the potential heterogeneity of PASC over different patient populations and the need for replication studies before robust conclusions about PASC can be made."

Paragraph in the discussion: "We observed clear heterogeneity after replicating...."

Response: Thanks for the great comments. We have toned down these sentences or made them more specific as suggested. By stringent screen criteria and extensive sensitivity analyses (e.g., controlling for the population size) as we discussed in the response to the previous comments, we hope to maximally reduce the probability that our primary findings in terms of the heterogeneity and commonality in the two cohorts were chance findings.

The presentation of the findings in Figures 2 and 3 (and the results and discussion sections) seem to indicate confusion between two possible approaches to interpreting the results across the two cohorts. It is common (for example in analyses of genome-wide association studies) to consider the first dataset as "discovery" and the second as "replication", with associations meeting significance criteria in the discovery dataset only considered confirmed when replicated in the second dataset. The following statement (lines 40-42) implies that this is what the authors intend: "... conditions including dementia, hair loss, pressure ulcers, pulmonary fibrosis, dyspnea, pulmonary embolism, chest pain, abnormal heartbeat, malaise, and fatigue, were replicated across two populations." However, they do not seem to interpret Figure 3 as demonstrating lack of replication for most conditions, and their description of the INSIGHT results implies that they consider all these to be confirmed in light of meeting the p value threshold within INSIGHT. It is unclear what they believe about the OneFlorida associations, which are discussed much more briefly. Similarly, the second paragraph of the discussion seems to focus on the INSIGHT results, although it does not mention which dataset results are being described.

Alternatively, the authors may be interested in (line 30) "the heterogeneous risks of PASC in different populations". In this case, they should present the results that met the p value threshold from each dataset, instead of focussing on the INSIGHT results (Figure 2, lines 150 to 210) and then comparing the results for post-acute sequelae identified in INSIGHT with those from OneFlorida. In any case, Figure 3 reproduces the INSIGHT results presented in Figure 2: it makes little sense to include the same INSIGHT results for post-acute sequelae in each figure. It is unclear why the authors compared results for post-acute sequelae but not medications between the cohorts.

Response: Thanks for the comments. We would like to do the following clarifications.

First, we tried to generate PASC signals from INSIGHT and OneFlorida+, and then to show their heterogeneity and commonality through comparisons. We started with the INSIGHT results as shown in Fig. 2, followed by the OneFlorida+ results and their comparisons as summarized in Fig. 3.

Second, we have revised the Section - Results from the OneFlorida+ Cohort and the comparison with results from INSIGHT, to further describe the difference between the two cohorts and to highlight the replicated results. We included the INSIGHT results in Fig. 3 for the convenience of comparison. We also compared medications and summarized the replicated medications in Extended Data Fig. 3. We hope these comparisons between two large EHR cohorts, covering both the heterogeneity and commonality, can provide new insights for PASC.

2. Subgroup analyses.

In their revised description of the subgroup analyses the authors focus on comparing burdens between subgroups. However, relative effects that are constant between a higher and lower burden setting will lead to a larger number of excess cases in the higher than lower burden setting. The authors should comment on how relative effects, as well as excess burdens, vary between subgroups.

Response: Thanks for the great comments. Indeed, your statement is exactly one of the reasons why we used excess burden (difference) in the subgroup analysis, aiming to provide another view of the relative effect (ratio in the primary analysis). Another reason is to be comparable with existing literature conducted on the VA cohort. ¹

We further clarified this point in the stratified analysis section and provided results in terms of adjusted hazard ratios in Extended Data Fig 12 and 13.

3. Different waves.

The evolution from ancestral to alpha to delta to omicron waves was accompanied by the availability of primary course vaccination and then booster vaccination, which provided successively greater protection against severe COVID-19. This should be accounted for when comparing the excess burden of PASCs over different waves.

Response: Thanks for the comments.

We have conducted a sensitivity analysis by adjusting for additional baseline vaccination status, including fully vaccinated, partially vaccinated, and no evidence of vaccination (results in the sensitivity analysis section, population size in Extended Data Table 4, and aHRs in Extended Data Fig. 8). We defined the fully vaccinated as two shots of mRNA vaccine (Pfizer, or Moderna) or one shot of J&J, according to the CDC guideline <https://www.cdc.gov/coronavirus/2019-ncov/vaccines/stay-up-to-date.html>. As shown in Extended Data Fig. 8, adjusting for these baseline vaccination covariates had little impact on the adjusted hazard ratios of selected PASC conditions compared with our primary analysis. We discuss the potential reasons as follows.

First, the vaccination began in early December 2020 (see the revised extended data fig 1), and more than half of the patients in our study cohorts got infected before the vaccine was available. Second, even for patients who got infected after December 2020, the portion of them who had any vaccination records was small (see Extended Data Table 4). Taking the INSIGHT NYC cohort as an example, the fully or partially vaccinated patients only accounted for 4.2% of the total population, which could barely impact the statistical conclusions of the screened conditions considering their significant hazard ratio. On the other hand, though 95.9% of patients had no evidence of vaccination, we cannot ascertain if they were not

vaccinated or due to missingness. We acknowledged this as a limitation of the EHR-based study, and one of our ongoing efforts is to link EHR with the vaccine registry databases to better understand how vaccination influences the risk of long COVID.

4. Causal inference

The authors say that they have “deleted all claims about “causal inference” to avoid confusion as our study is a retrospective analysis of observational EHR cohorts.” However, in various places they continue to refer to having used a target trial approach. Since the use of target trials explicitly aims to make causal inferences, references to target trials should also be deleted.

Response: Thanks for the comments. We have further removed “target trial” and associated references from the main text. Again, we identified our work as a hypothesis generation work using RWDs and adjusted analysis rather than claiming causal associations.

5. Inverse probability weighting.

Throughout, the authors refer to “inverse propensity score weighting”. Although the weights are based on the propensity score, they are not the inverse of the propensity score. They are inverse probability weights where the probability is that of being a case, as is made clear in the formula provided in the methods section. The wording should be changed throughout the manuscript.

Response: Thanks for the comments. We have changed the wording throughout the manuscript by using the term Inverse probability of treatment weighting (IPTW), which is a standard term used in prior literature when referring to this type of method.^{2,3}

The authors have trimmed the stabilised weights in order to avoid instability in estimates. However, this corresponds to not fully controlling the confounding. I would have preferred them to conduct a sensitivity analysis conditioning on a nonlinear function of the propensity score, which avoids this problem, but they did not take up this suggestion. There isn't much advantage to IPW when we are only dealing with baseline confounding.

Response: Thanks for the suggestion. We have conducted this additional sensitivity analysis and summarized the results in the Sensitivity analysis section and Extended Data Fig. 10. Specifically, we used a gradient-boosting decision tree model to estimate the propensity score, replicated the analyses on the two cohorts, and compared the aHR in our primary analysis. As shown in Extended Data Fig. 10, we found similar aHRs and got the same set of replicated PASC conditions as we observed in the primary analysis.

In addition, we also conducted the negative outcome control using multiple outcomes in both databases (Extended Data Table 1) to further reduce potential residual confounding.

Minor comments

Line 57: "... or result in biased findings". This is vague – either clarify or delete.

Response: Thanks for the comments. We have deleted it as suggested.

Lines 57-58: "... prior studies have typically been conducted on specific populations". There are quite a few general population studies and these should be cited.

Response: Thanks for the comments. We have revised the text with citations and shown that existing literature typically focused on one specific cohort and didn't compare results across different populations.

Typo in the paragraph on sensitivity analysis: "potentially significant PASC diagnoses in the INSIGHT the OneFlorida+".

Response: Thanks for the comment. We have corrected it in the text.

Methods: "examined a total of 596 incident diagnoses (Supplemental Table 2) and medication use...". Should this be "

Methods: "In addition, periods (March 2020 – June 2020, July 2020 – October 2020, November 2020 - February 2021, March 2021 – June 2021, July 2021 – November 2021) of the index date were used to account for potentially different stages of the pandemic." Given the rapid changes in incidence of COVID with calendar time this may not sufficiently account for confounding by calendar time. Instead, use a continuous function of calendar time such as a cubic spline.

Response: Thanks for the suggestion. We have added another set of sensitivity analyses in the Sensitivity analysis section, using the cubic B-spline to model the dates in terms of the number of days since March 2020. We replicated our analyses on two datasets, and we found consistent results as shown in Extended Data Fig. 9.

In addition, as shown in Extended Data Table 1., the negative outcome control using multiple outcomes in both two databases further reduced the potential impact of residual confounding on the primary results.

Reviewer #3 (Remarks to the Author):

I am happy with the changes made by the authors in relation to my previous comments, and support the publication of this work.

Reference

1. Xie, Y., Bowe, B. & Al-Aly, Z. Burdens of post-acute sequelae of COVID-19 by severity of acute infection, demographics and health status. *Nat Commun* **12**, 6571 (2021).
2. Robins, J. M., Hernán, M. Á. & Brumback, B. Marginal Structural Models and Causal Inference in Epidemiology. *Epidemiology* **11**, 550–560 (2000).
3. Austin, P. C. & Stuart, E. A. Moving towards best practice when using inverse probability of treatment weighting (IPTW) using the propensity score to estimate causal treatment effects in observational studies. *Statistics in Medicine* **34**, 3661–3679 (2015).

REVIEWERS' COMMENTS

Reviewer #2 (Remarks to the Author):

The authors have further changed their manuscript following my second review, and I am grateful for the extra work that they have done in conducting the requested sensitivity analyses and addressing whether the different sample sizes explained the different findings between the two cohorts.

However, the authors have not amended Figures 2 and 3 in the way that I suggested. They continue to present identical results from INSIGHT in Figures 2 and 3, while not presenting all of the results from OneFlorida in the main figures. I continue to believe that it would be better to present all the findings for incident diagnoses for both INSIGHT and OneFlorida in Fig 2 (INSIGHT on the left and OneFlorida on the right), and use Figure 3 in the same way to present all the findings for incidence prescriptions of medications. This would avoid the need for Extended Figure 3. The authors' response reiterates what they did (and continue to do) but does not convincingly justify the priority given to INSIGHT over OneFlorida, given that they emphasise heterogeneity between the cohorts rather than regarding OneFlorida as a replication/validation dataset. The non-replication of many findings is an important result of the paper, particularly since the authors have excluded differences in sample size as an explanation. In their discussion of this issue (which could be further extended) the authors compare characteristics of the two cohorts, and suggest that differences in age and social disadvantage might be relevant. In this context, they say that "OneFlorida+ patients might be less likely to present for care during a relatively short post-acute phase". I may have missed it, but if they do not already report the median and IQR length of follow up in the two cohorts they should do so. If these differ, a further sensitivity analysis restricted to a comparable follow up period may be appropriate.

REVIEWERS' COMMENTS – Round 3

Reviewer #2 (Remarks to the Author):

The authors have further changed their manuscript following my second review, and I am grateful for the extra work that they have done in conducting the requested sensitivity analyses and addressing whether the different sample sizes explained the different findings between the two cohorts.

However, the authors have not amended Figures 2 and 3 in the way that I suggested. They continue to present identical results from INSIGHT in Figures 2 and 3, while not presenting all of the results from OneFlorida in the main figures. **I continue to believe that it would be better to present all the findings for incident diagnoses for both INSIGHT and OneFlorida in Fig 2 (INSIGHT on the left and OneFlorida on the right), and use Figure 3 in the same way to present all the findings for incidence prescriptions of medications.** This would avoid the need for Extended Figure 3. The authors' response reiterates what they did (and continue to do) but does not convincingly justify the priority given to INSIGHT over OneFlorida, given that they emphasise heterogeneity between the cohorts rather than regarding OneFlorida as a replication/validation dataset. The non-replication of many findings is an important result of the paper, particularly since the authors have excluded differences in sample size as an explanation. In their discussion of this issue (which could be further extended) the authors compare characteristics of the two cohorts, and suggest that differences in age and social disadvantage might be relevant. In this context, they say that "OneFlorida+ patients might be less likely to present for care during a relatively short post-acute phase". I may have missed it, but if they do not already report **the median and IQR length of follow up in the two cohorts they should do so.** If these differ, a further sensitivity analysis restricted to a comparable follow up period may be appropriate.

Response: thanks for your great suggestions. First, we revised Figure 2, as you suggested, to present all the findings from INSIGHT and OneFlorida+. We summarized diagnoses and medications from two cohorts in Fig 2, aiming to provide a holistic view of comparisons between the two cohorts. Second, we added the median and IQR of follow-up days to Table 1, which were comparable between exposure groups.

Characteristics	INSIGHT			OneFlorida+		
	SARS-CoV-2 Positive (N=35,275)	SARS-CoV-2 Negative (N=326,126)	SMD ^b	SARS-CoV-2 Positive (N=22,341)	SARS-CoV-2 Negative (N=177,010)	SMD ^b
Follow-up days (IQR)	258 (163-418)	269 (145-388)	0.09	207 (109-367)	250 (122-409)	-0.17

Thanks again for your suggestions and comments.